# The rapid developmental rise of somatic inhibition disengages hippocampal dynamics from self-motion

Robin F Dard[1], Erwan Leprince[1], Julien Denis[1], Shrisha Rao Balappa[2], Dmitrii Suchkov[1], Richard Boyce[1], Catherine Lopez[1], Marie Giorgi-Kurz[1], Tom Szwagier[1,3], Théo Dumont[1,3], Hervé Rouault[2], Marat Minlebaev[1], Agnès Baude[1], Rosa Cossart[1]*, Michel A Picardo[1]*

[1]Turing Centre for Living systems, Aix-Marseille University, INSERM, INMED U1249, Marseille, France; [2]Turing Centre for Living systems, Aix-Marseille University, Université de Toulon, CNRS, CPT (UMR 7332), Marseille, France; [3]Mines ParisTech, PSL Research University, Paris, France

*For correspondence:
rosa.cossart@inserm.fr (RC);
michel.picardo@inserm.fr (MAP)

Competing interest: The authors declare that no competing interests exist.

**Abstract** Early electrophysiological brain oscillations recorded in preterm babies and newborn rodents are initially mostly driven by bottom-up sensorimotor activity and only later can detach from external inputs. This is a hallmark of most developing brain areas, including the hippocampus, which, in the adult brain, functions in integrating external inputs onto internal dynamics. Such developmental disengagement from external inputs is likely a fundamental step for the proper development of cognitive internal models. Despite its importance, the developmental timeline and circuit basis for this disengagement remain unknown. To address this issue, we have investigated the daily evolution of CA1 dynamics and underlying circuits during the first two postnatal weeks of mouse development using two-photon calcium imaging in non-anesthetized pups. We show that the first postnatal week ends with an abrupt shift in the representation of self-motion in CA1. Indeed, most CA1 pyramidal cells switch from activated to inhibited by self-generated movements at the end of the first postnatal week, whereas the majority of GABAergic neurons remain positively modulated throughout this period. This rapid switch occurs within 2 days and follows the rapid anatomical and functional surge of local somatic GABAergic innervation. The observed change in dynamics is consistent with a two-population model undergoing a strengthening of inhibition. We propose that this abrupt developmental transition inaugurates the emergence of internal hippocampal dynamics.

## Editor's evaluation

This study investigates hippocampal dynamics over the course of early postnatal development with respect to spontaneous movements. Pioneering in vivo imaging in the hippocampus of neonatal mice, the authors find evidence for an abrupt developmental transition in this neural activity at the end of the first postnatal week in rodents and contributes to understanding how cognitive functions could emerge from the immature brain.

## Introduction

The adult hippocampus serves multiple cognitive functions, including navigation and memory. These functions rely on the ability of hippocampal circuits to integrate external inputs conveying multisensory, proprioceptive, contextual, and emotional information onto internally generated dynamics. Therefore, the capacity to produce internally coordinated neuronal activity detached from environmental inputs

is central to the cognitive functions of the hippocampus such as planning and memory (*Buzsáki, 2015*; *Buzsáki and Moser, 2013*). In contrast to the adult situation, the developing hippocampus, like many developing cortical structures, is mainly driven by bottom-up external environmental and body-derived signals, including motor twitches generated in the spinal cord and/or the brainstem (*Dooley et al., 2020*; *Inácio et al., 2016*; *Karlsson et al., 2006*; *Mohns and Blumberg, 2010*; *Del Rio-Bermudez et al., 2020*; *Valeeva et al., 2019a*). These produce early sharp waves (eSW) conveyed by inputs from the entorhinal cortex (*Valeeva et al., 2019a*). The emergence of self-organized sequences without reliance on external cues in the form of sharp wave ripples (SWRs) is only observed after the end of the second postnatal week and sequential reactivations even a week later (*Farooq and Dragoi, 2019*; *Muessig et al., 2019*). Therefore, early hippocampal activity as measured with electrophysiological recordings is first externally driven while the emergence of internal dynamics is protracted. The timing and the circuit mechanisms of the switch between motion-guided and internally produced hippocampal dynamics remain unknown. They have been proposed to rely on the maturation of CA3 and extrinsic hippocampal inputs; however, a possible role of local connectivity, in particular, recurrent somatic inhibition, cannot be excluded (*Cossart and Khazipov, 2022*).

Local GABAergic interneurons could be critically involved in this phenomenon for several reasons. First, both theoretical and experimental work suggest that self-organized internal neuronal network dynamics require feedback connections to produce an emergent state of activity independently from the incoming input (*Hopfield and Tank, 2005*; *Hopfield, 1982*; *Yuste, 2015*). Feedback circuits are mainly GABAergic in CA1 (but not necessarily inhibitory), given the scarcity of recurrent glutamatergic connections in that hippocampal subregion (*Bezaire and Soltesz, 2013*). Second, GABAergic interneurons, in particular, the perisomatic subtypes, are long known to shape the spatial and temporal organization of internal CA1 dynamics (*Buzsáki, 2015*; *Lee et al., 2014*; *Soltesz and Losonczy, 2018*; *Valero et al., 2015*). However, GABAergic perisomatic cells display a delayed maturation profile both at structural (*Jiang et al., 2001*; *Marty et al., 2002*; *Morozov and Freund, 2003*; *Tyzio et al., 1999*) and functional levels (*Ben-Ari, 2002*; *Doischer et al., 2008*; *Jiang et al., 2001*; *Khazipov et al., 2004*; *Marty et al., 2002*; *Morozov and Freund, 2003*; *Murata and Colonnese, 2020*; *Tyzio et al., 1999*), and the precise developmental timeline for their postnatal development remains unknown, partly due to the difficulty in labeling them (*Donato et al., 2017*).

Here, we investigate the evolution of CA1 dynamics during the first and second postnatal weeks of mouse development with an eye on the specific patterning of activity of CA1 GABAergic neurons. To this aim, we adapted two-photon calcium imaging of CA1 dynamics using virally expressed GCaMP6 through a cranial window in non-anesthetized pups. We show that the first postnatal week ends with an abrupt switch in the representation of self-motion in CA1: principal neurons were synchronized by spontaneous movement before P9, whereas self-motion decreased their activity after that time point. Consistent with a two-population neuronal model, this switch was locally paralleled by the rapid anatomical and functional surge of somatic GABAergic interneurons and no significant change in external inputs. Self-generated bottom-up inputs may thus directly contribute to the emergence of somatic GABAergic inhibition and in this way calibrate local circuits to the magnitude of external inputs prior to the opening of experience-dependent plasticity.

## Results

### Progressive evolution of CA1 neuronal dynamics

In order to induce stable and early expression of the calcium indicator protein GCaMP6s, pups were injected with the AAV1-hSyn-GCaMP6s.WPRE.SV40 virus in the brain lateral ventricle on the day of birth (P0, *Figure 1A*, *Figure 1—figure supplement 1A*). Five to twelve days after injection, the hippocampal CA1 region of non-anesthetized pups was imaged through a window implant placed on the same day (*Figure 1A*, see 'Materials and methods'). We first quantified the amount of sleep/wake cycle in P5–6 mice after cranial window surgery and electromyogram (EMG) nuchal electrodes implantation. We found that mice spent 74% (±6) of their time in active sleep (*Figure 1—figure supplement 1B*), which is comparable to previous reports (*Jouvet-Mounier et al., 1970*). This indicates that the window implant did not alter this characteristic of behavior in early postnatal stages. In the same way, the acute window implant did not significantly alter electrophysiological network patterns. These were measured using in vivo bilateral silicon probes recordings of eSW (*Figure 1—figure supplement*

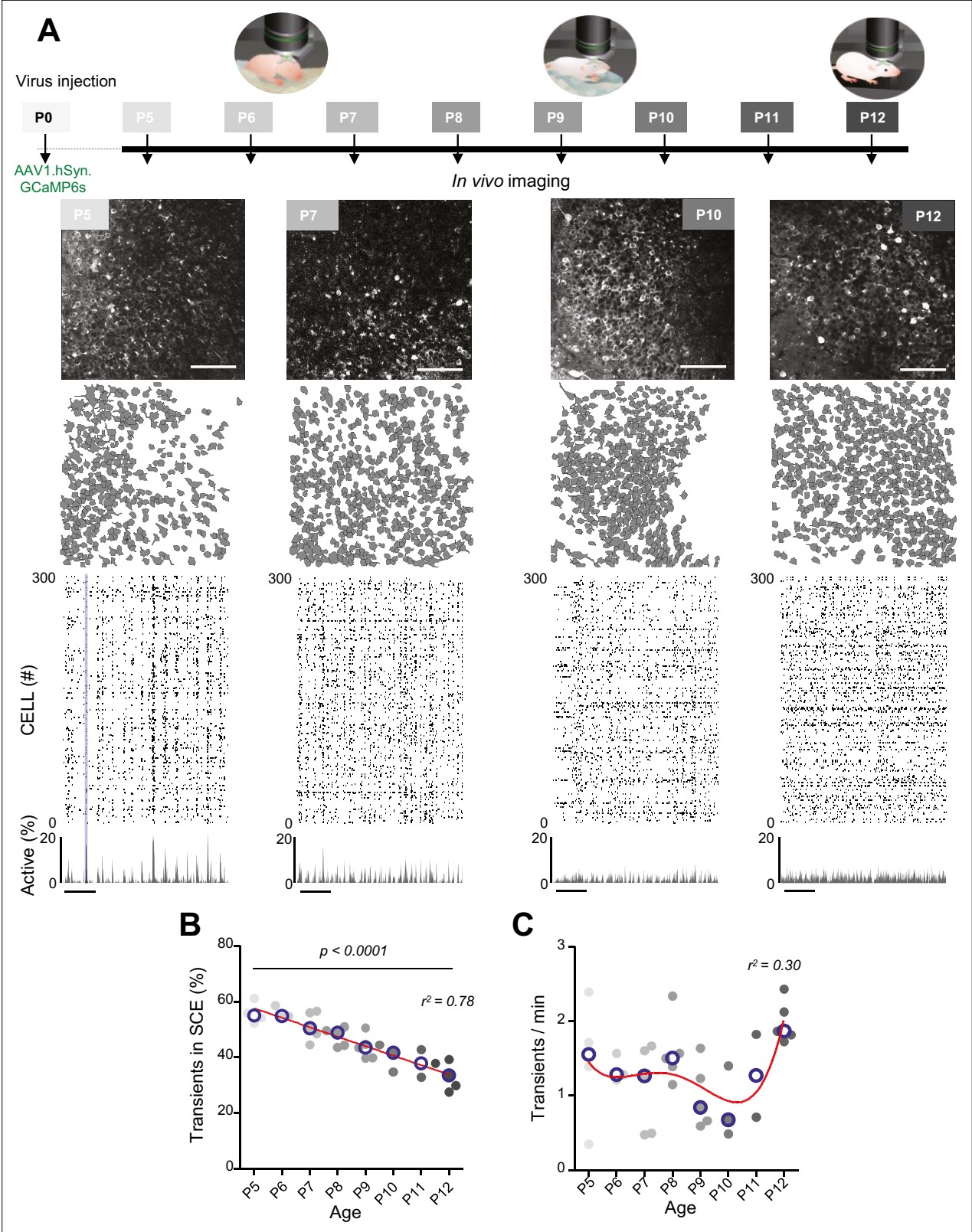

**Figure 1.** Evolution of CA1 dynamics during the first two postnatal weeks. (**A**) Schematic of the experimental timeline. On postnatal day 0 (P0), 2 μL of a nondiluted viral solution was injected into the left lateral ventricle of mouse pups. From 5 to 12 days after injection (P5–P12), acute surgery for window implantation above the corpus callosum was performed and followed by two-photon calcium imaging recordings. Top panel: four example recordings are shown to illustrate the imaging fields of view in the *stratum pyramidale* of the CA1 region of the hippocampus (scale bar: 100 μm). Middle

*Figure 1 continued on next page*

*Figure 1 continued*

panel: contour maps showing the cells detected using Suite2p in the corresponding fields of view. Bottom panel: raster plots inferred by DeepCINAC activity classifier, showing 300 randomly selected cells over the first 5 min of recording obtained for these imaging sessions (P5, P7, P10, and P12, see full raster plots for these imaging sessions in *Figure 1—figure supplement 1D*). In the raster plot from the P5 mouse, the blue rectangle illustrates one synchronous calcium event (SCE). Scale bar for time is 1 min. Calcium Imaging Complete Automated Data Analysis (CICADA) configuration files to reproduce example rater plots and cell contours are available in *Figure 1—source data 1*. (**B**) Evolution of the ratio of calcium transients within SCEs over the total number of transients across age. Each dot represents a mouse pup and is color coded from light gray (P5) to black (P12), the open blue circles represent the median of the age group. The red line represents the linear fit of the data with $r^2 = 0.78$, p<0.0001 (N = 32 pups). Results to build the distribution, as well as CICADA configuration file to reproduce the analysis, are available in *Figure 1—source data 1*. (**C**) Evolution of the number of transients per minute across age. Each dot represents the mean transient frequency from all cells imaged in one animal and is color coded from light gray (P5) to black (P12). The red line represents the nonlinear fit (fourth-order polynomial, least-squares method) of the data with $r^2 = 0.30$ (N = 32 pups). The open blue circles represent the median of the age group. Results to build the distribution, as well as CICADA configuration file to reproduce the analysis, are available in *Figure 1—source data 1*.

The online version of this article includes the following source data and figure supplement(s) for figure 1:

**Source data 1.** Analysis configuration files and numerical data used in *Figure 1*.

**Figure supplement 1.** Impact of acute window implant on CA1 dynamics.

*1C*, see 'Discussion') in P6–8 (n = 4) and P11 (n = 2) pups expressing GCaMP6s with a frequency of 2.6 eSW/min (25% 1.15 eSW/min and 75% 4.16 eSW/min) for the ipsilateral side and 3.49 eSW/ min (25% 1.96 eSW/min and 75% 5.1 eSW/min, p-value=0.39) for the contralateral side (*Figure 1— figure supplement 1C*). This slight but nonsignificant reduction in eSW frequency recorded from the ipsilateral hemisphere was similarly reported in a previous study using the same surgical approach (see 'Discussion', *Graf et al., 2021*). eSW synchronization between hemispheres was preserved (*Valeeva et al., 2019b*) but with a 12 ms delay between the two hemispheres (peak at 0.087 ± 0.027 s, *Figure 1—figure supplement 1C*), possibly explained by a slight drop in local temperature due to the chamber placement as described previously (*Reig et al., 2010*). Finally, we checked for the presence of other types of oscillations in both hemispheres and observed a peak in the theta range in the P11 mouse pups in both hemispheres (ipsi: peak at 4.3 Hz of amplitude $3.6 \times 10^{-3} \pm 1.2 \times 10^{-3}$ mV$^2$/Hz; contra: peak at 4.1 Hz of amplitude $2 \times 10^{-3} \pm 3 \times 10^{-4}$ mV$^2$/Hz (jackknife standard deviation), *Figure 1—figure supplement 1C*). In general, a slight increase in the peak power of most electrophysiological network oscillations (below 20 Hz) was observed (*Figure 1—figure supplement 1C*). Altogether, we can conclude that the presence of the window implant minimally disrupted the electrophysiological network patterns and sleep–wake cycle of developing rodents during that early postnatal period. Thus, we pursued the description of early multineuron CA1 dynamics using calcium imaging (62 imaging sessions, 35 mouse pups aged between 5 and 12 days, yielding a total of 33,412 cells, see *Supplementary file 1* for details of each session and their inclusion in the figures).

The contours of the imaged neurons and their calcium fluorescence events were extracted using Suite2P (*Pachitariu et al., 2017*) and DeepCINAC (*Denis et al., 2020*), respectively. Representative examples of fields of view, contour maps, and activity raster plots from recordings in P5, P7, P10, and P12 mouse pups are shown in *Figure 1A*. Neuronal activity was stable over the duration of the recording (*Figure 1—figure supplement 1D and E* – median change: 0.08 transients/minute, N = 31). Consistent with previous electrophysiological studies (*Leinekugel et al., 2002*; *Mohns and Blumberg, 2008*; *Valeeva et al., 2019a*), spontaneous neuronal activity in the CA1 region of P5–6 pups alternated between recurring population bursts (synchronous calcium events [SCEs]) and periods of low activity (*Figure 1A*). In P5–6 mouse pups, more than half of the detected calcium transients occurred within SCEs (P5: median value 55% N = 4, n = 8; P6: median value 55% N = 3, n = 5; N: mice, n: imaging sessions, *Figure 1B*). Activity then became progressively continuous as evidenced by the linear decrease in the proportion of calcium transients occurring during SCEs ($r^2 = 0.78$, p<0.0001) to finally reach 33% in P12 mouse pups (P12: N = 5, n = 8, *Figure 1B*). Reminiscent of a transient period of 'neural quiescence' at the beginning of the second postnatal week (*Domínguez et al., 2021*), we observed a nonlinear evolution in the cell activation frequency with a local minimum around P10 ($r^2 = 0.30$, *Figure 1C*). We conclude that CA1 dynamics progressively evolve from discontinuous to continuous during the first two postnatal weeks, in agreement with previous electrophysiological studies (*Cossart and Khazipov, 2022*; *Mohns and Blumberg, 2008*; *Valeeva et al., 2019a*).

## Early SCEs correlate with spontaneous motor activity

Previous extracellular electrophysiological recordings indicated that, in developing rodents, CA1 dynamics followed spontaneous motor activity during the first postnatal week (*Karlsson et al., 2006*; *Del Rio-Bermudez et al., 2020*; *Valeeva et al., 2019a*). Hence, we next examined the relationship between population activity and movement as monitored using either piezo recordings or infrared cameras (see 'Materials and methods'). Because our surgical procedure could potentially affect CA1 dynamics in response to contralateral movement, we computed peri-movement time histograms (PMTHs) by plotting the fraction of active neurons centered on the onset of all ipsi- or contralateral limbs movements. Both spontaneous limb movements were followed by an increase in CA1 activity (peak ipsi = 3.6%, peak contra = 2.8%, chance level 3.4%, *Figure 1—figure supplement 1E*), showing that the surgery was not preventing the hippocampal response to contralateral limb movements. Still, contralateral limb movements recruited a slightly lower fraction of active cells (see 'Discussion'). In mouse pups younger than P9, movements were followed by a significant increase in the percentage of active cells exceeding the chance level (*Figure 2A*; P5–8 median above chance level, *Videos 1–3*) and an increase in the average DF/F fluorescence signal (*Figure 2—figure supplement 1A*). In contrast, after P9, movements were followed by a significant decrease in activity below chance level (*Figure 2A*, P10–12 median below chance level, *Videos 4–6*) and a decreased DF/F fluorescence signal (*Figure 2—figure supplement 1A*). Short myoclonic movements such as twitches, happening during periods of active sleep (*Gramsbergen et al., 1970*; *Jouvet-Mounier and Astic, 1968*; *Karlsson et al., 2006*) as opposed to longer movements, happening mostly during wakefulness, may induce different activity patterns in the hippocampus (*Mohns and Blumberg, 2008*). This difference between wake movements and active-sleep twitches during development is proposed to rely on a gating of sensory feedback associated with movement during wake (*Dooley and Blumberg, 2018*; *Tiriac and Blumberg, 2016*). Accordingly, when combining calcium imaging with nuchal EMG recordings in one P5 mouse pup, we observed an increase in the percentage of active cells and in the DF/F fluorescence signal following movements occurring both during REM sleep and wakefulness (*Figure 2—figure supplement 1C*). However, when combining all mouse pups, and considering separately twitches (occurring during REM/active sleep) and complex movements (occurring during wakefulness), based on video recordings, we found that the two movement types did not significantly differ in their impact on CA1 activity (*Figure 2—figure supplement 1B*). Given this lack of difference, all movement types were thus combined in the following analysis steps. Post-movement activity was next computed, as defined by the number of active cells in the 2 s following movement onset divided by the number of active cells within a 4-s-long time window centered on movement onset (see 'Materials and methods,' *Figure 2B*, and *Figure 2—figure supplement 1D*). The median post-movement activity progressively decreased from P5 to P9 (mean difference between consecutive age groups of 3.1 ± 0.7%) until it suddenly dropped at P10 (–13.5% between P9 and P10) and stabilized until P12. P9 marked the transition in the relationship between movement and CA1 activity. Indeed, the median post-movement activity exceeded 50% from P5 to P8 (P5: 71%; P6: 65%; P7: 60%; P8: 56%). This is consistent with the evolution of PMTHs (*Figure 2A*). After P9, the median post-movement activity was lower than 50% (P10: 39%; P11: 35%; P12: 40%), thus revealing the inhibitory action of movement on activity. We next defined as 'inhibiting movements' all the movements with a post-movement activity lower than 40% and computed their proportion in each mouse (*Figure 2C*). The proportion of 'inhibiting' movements was stable before P9 (P5: 11%; P6: 16%; P7: 10%, P8: 15%). Again, P9 marks a transition since we observed that approximately half of the movements were followed by an inhibition of CA1 activity in P10–12 mice (P10: 55%; P11: 58%; P12: 48%). The proportion of 'inhibiting' movements varies with age as a sigmoid function with P9 being the transition time point (V50 = 9.015, $r^2$ = 0.75). In line with the emergence of movement-induced inhibition, the fraction of neurons significantly associated with immobility also increased with age, also following a sigmoidal function (*Figure 2D*, sigmoid fit V50 = 9.022, $r^2$ = 0.55). Altogether, these results indicate that the end of the first postnatal week marks a transition in the evolution of CA1 dynamics, with both a decorrelation and a 'detachment' of neuronal activity from spontaneous motor activity. We next investigated the circuit mechanisms supporting these changes.

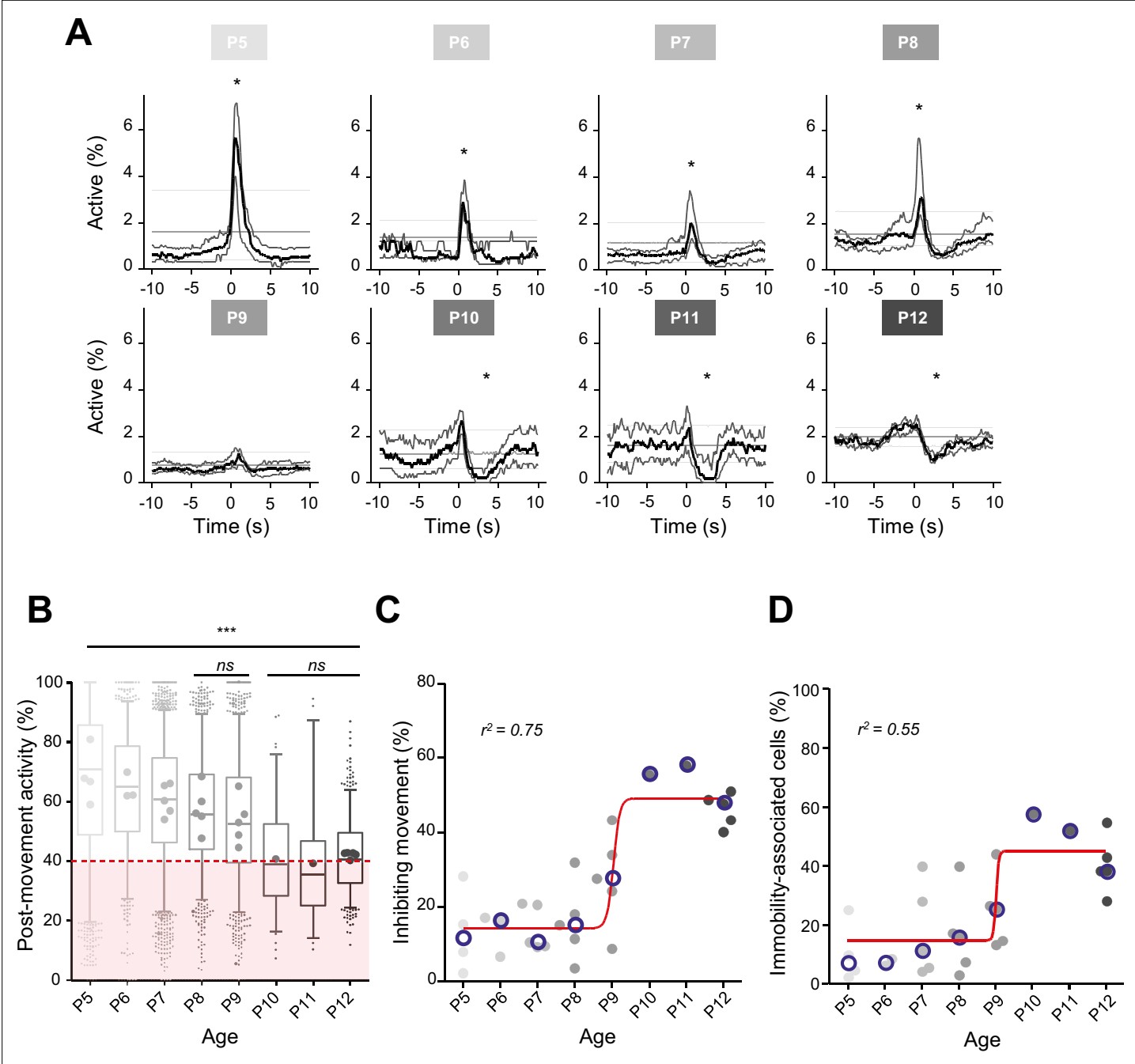

**Figure 2.** Linking CA1 dynamics to movement during the first two postnatal weeks. (**A**) Peri-movement time histograms (PMTH) representing the percentage of active cells centered on the onset of the mouse movements. The dark line indicates the median value, and the two thick gray lines represent the 25th and 75th percentiles from the distribution made of all median PMTHs from the sessions included in the group. Overall are included: P5: N = 4, n = 8; P6: N = 3, n = 5; P7: N = 5, n = 12; P8: N = 5, n = 8; P9: N = 5, n = 11; P10: N = 1, n = 1; P11: N = 1, n = 1; P12: N = 5, n = 7 (N, number of mice; n, number of imaging sessions). In all panels, the thin straight gray lines represent the 5th percentile, the median, and the 95th percentile of the distribution made of all median PMTHs resulting from surrogate raster plots from the sessions included in the group. Black asterisk indicate that the median value is above the 95th percentile or below the 5th percentile from the surrogates. Results to build the PMTH, as well as Calcium Imaging Complete Automated Data Analysis (CICADA) configuration file to reproduce the analysis, are available in *Figure 2—source data 1*. (**B**) Distribution of post-movement activity across age. Each box plot is built from all detected movements for the given age group. Whiskers represent the 5th and 95th percentiles with post-movement activity falling above or below represented as small dots. The average post-movement activity observed for each mouse pup is represented by the large dots color coded from light gray (P5) to black (P12). The red area illustrates the movement falling in the category of 'inhibiting' movements. P5: four mice, 1519 movements; P6: three mice, 766 movements; P7: five mice, 2067 movements; P8: five mice, 1105 movements; P9: five mice, 1272 movements; P10: one mouse, 83 movements; P11: one mouse, 57 movements; P12: three mice, 493 movements. Global

*Figure 2 continued on next page*

*Figure 2 continued*

effect of age was found significant (ANOVA, eight groups, *F* = 107.7, p-value<0.0001). Comparison between age groups shows that except all three possible pairs made of P10–P11–P12 and the P8–P9 pair, all pairs were significantly different (p-value<0.005, post hoc Bonferroni's multiple-comparison test). Results to build the distributions, as well as Calcium Imaging Complete Automated Data Analysis (CICADA) configuration file to reproduce the analysis, are available in *Figure 2—source data 1*. (**C**) Distribution of the proportion of 'inhibiting' movements across age. Each dot represents a mouse pup and is color coded from light gray (P5) to black (P12). The open blue circles represent the median of the age group. The red line shows a sigmoidal fit with V50 = 9.015, $r^2$ = 0.75 (least-squares method). Results to build the distribution, as well as CICADA configuration file to reproduce the analysis, are available in *Figure 2—source data 1*. (**D**) Distribution of the proportion of significantly immobility-associated cells as a function of age. Each dot represents a mouse and is color coded from light gray (P5) to black (P12). The open blue circles represent the median of the age group. The red line shows a sigmoidal fit with V50 = 9.022, $r^2$ = 0.55 (least-squares method). Results to build the distribution, as well as CICADA configuration file to reproduce the analysis, are available in *Figure 2—source data 1*.

The online version of this article includes the following source data and figure supplement(s) for figure 2:

**Source data 1.** Analysis configuration files and numerical data used in *Figure 2*.

**Figure supplement 1.** Linking CA1 dynamics to different movements and brain states.

## GABAergic neurons remain activated by spontaneous movement throughout the first two postnatal weeks

As a first step to identifying the circuit mechanisms for this switch, we focused on local circuits and disentangled the respective contribution of local GABAergic neurons and principal cells to CA1 dynamics as well as their relation to movement. To this aim, we identified GABAergic neurons with the expression of a red reporter (tdTomato) in *GAD1^{Cre/+}* pups virally infected with AAV9-FLEX-CAG-tdTomato and AAV1.hSyn.GCaMP6s (*Figure 3A and B*, *Figure 3—figure supplement 1A*). In addition, we used these imaging experiments (*Figure 3—figure supplement 1B*, top row) to train a cell classifier inferring interneurons in the absence of any reporter (*Figure 3—figure supplement 1B*, bottom row). This classifier was able to infer interneurons with 91% precision (*Figure 3—figure supplement 1C*; *Denis et al., 2020*). 'Labeled' and 'inferred' GABAergic neurons were combined into a single group referred to as 'interneurons'' in the following (*Figure 3—figure supplement 1D and E*). As illustrated in a representative raster plot from a P5 mouse, both pyramidal cells (black) and interneurons (red) were activated during movement (vertical gray lines, *Figure 3A*). This was confirmed when computing the PMTH for pups aged less than P9, with the activation of the two neuronal populations after movement exceeding chance level (P5–8: N = 17, n = 33, pyramidal cells: baseline value = 0.51%, peak value = 2.1%, interneurons: baseline value = 2.1%, peak value 7.9%, N: number of mice, n: number of imaging sessions, *Figure 3A*, *Figure 3—figure supplement 2A*). In line with the above results (*Figure 2A*),

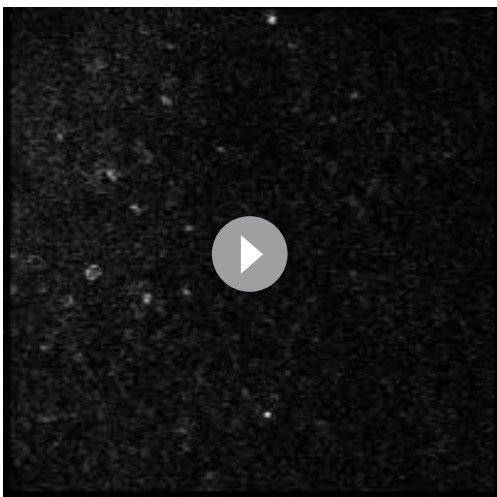

**Video 1.** First example of calcium imaging movies from P5 mouse pups centered on the onset of a twitch. The twitch is indicated by T in the upper-left corner of the movie. Imaging 2× speed up.

https://elifesciences.org/articles/78116/figures#video1

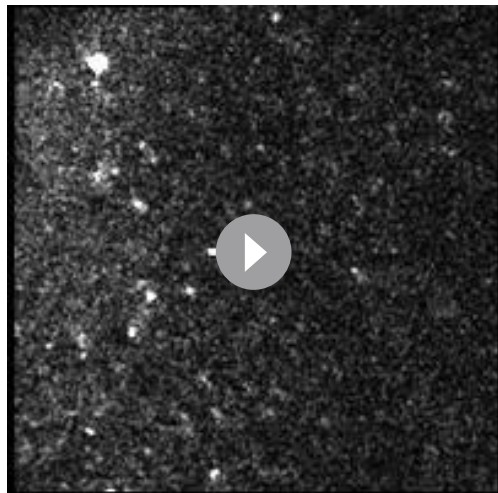

**Video 2.** Second example of calcium imaging movies from P5 mouse pups centered on the onset of a twitch. The twitch is indicated by T in the upper-left corner of the movie. Imaging 2× speed up.

https://elifesciences.org/articles/78116/figures#video2

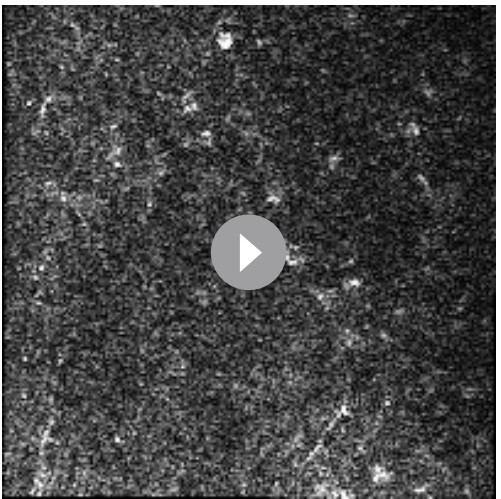

**Video 3.** Third example of calcium imaging movies from P5 mouse pups centered on the onset of a twitch. The twitch is indicated by T in the upper-left corner of the movie. Imaging 2× speed up.

https://elifesciences.org/articles/78116/figures#video3

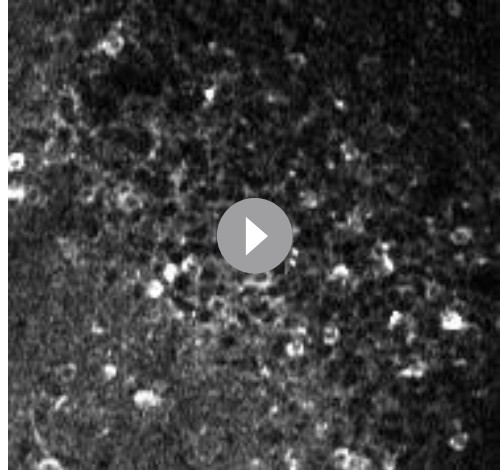

**Video 4.** First example of calcium imaging movies from P12 mouse pups centered on the onset of a complex movement. The complex movement is indicated by M in the upper-left corner of the movie. Imaging 2× speed up.

https://elifesciences.org/articles/78116/figures#video4

pups older than P9 showed a significant reduction (below chance level) in the proportion of active pyramidal cells following movement (*Figure 3B* P10–12: N = 7, n = 9, baseline value = 1.3%, trough value = 0.4%). In contrast, interneurons remained significantly activated following movement even past P9 (P10–12: N = 7, n = 9, baseline value = 3.9%, peak value = 10%, *Figure 3B*, *Figure 3—figure supplement 2B*). We conclude that the link between movement and activity evolves differentially toward the start of the second postnatal week when comparing pyramidal neurons and GABAergic interneurons, the former being inhibited or detached from movements while the latter remaining activated. This suggests that pyramidal neurons could be directly inhibited by local interneurons after the first postnatal week, following a functional maturation of GABAergic outputs onto principal cells. Alternatively, this could result from differential changes in

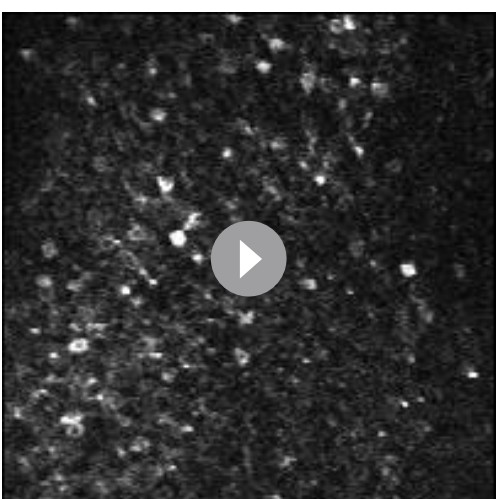

**Video 5.** Second example of calcium imaging movies from P12 mouse pups centered on the onset of a complex movement. The complex movement is indicated by M in the upper-left corner of the movie. Imaging 2× speed up.

https://elifesciences.org/articles/78116/figures#video5

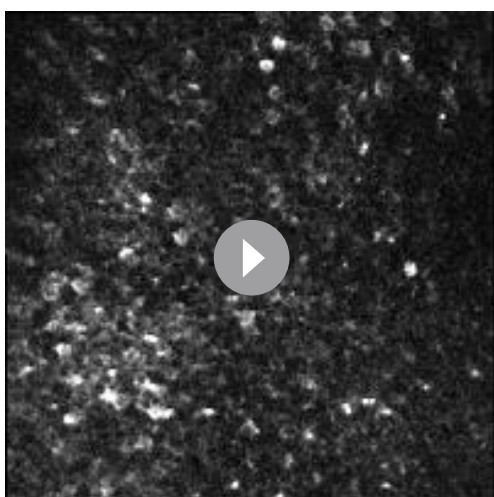

**Video 6.** Third example of calcium imaging movies from P12 mouse pups centered on the onset of a complex movement. The complex movement is indicated by M in the upper-left corner of the movie. Imaging 2× speed up.

https://elifesciences.org/articles/78116/figures#video6

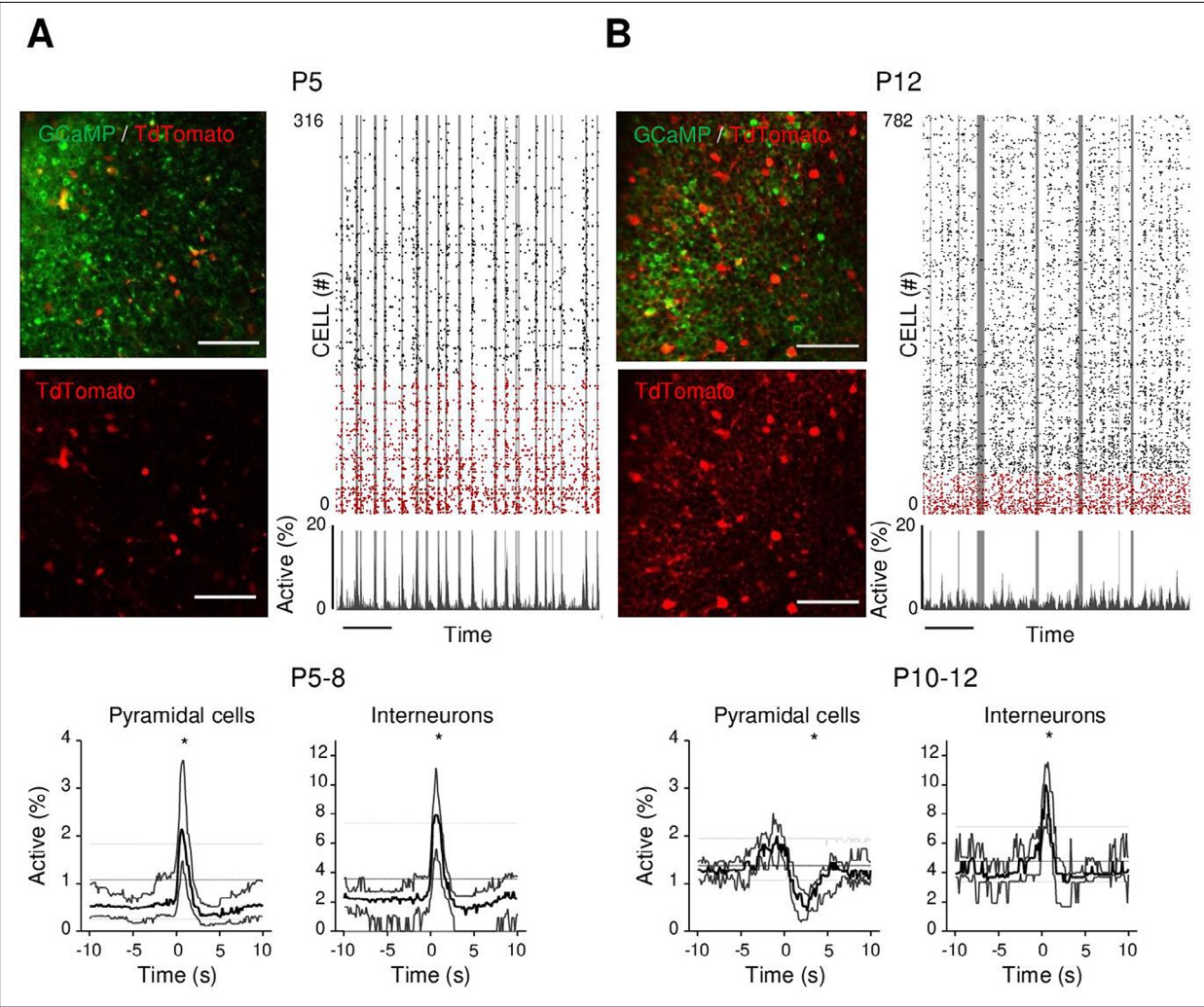

**Figure 3.** Differential recruitment of CA1 glutamatergic and GABAergic neurons. (**A**). Top panel: imaged field of view and associated raster plot from an example imaging session in the *stratum pyramidale* from one P5 *Gad1^Cre/+* mouse pup (scale bar = 100 μm). Imaged neurons expressed GCaMP6s. Interneurons were identified by the Cre-dependent expression of the red reporter tdTomato. In the raster plot, neurons are sorted according to their identification as pyramidal cells (black) or interneurons (red), vertical gray lines indicate movements of the mouse. Scale bar: 60 s. Bottom panel: peri-movement time histograms (PMTHs) for pyramidal cells and interneurons combining all imaging sessions from mice aged between P5 and P8 (N = 17 mice, n = 33 imaging sessions). The dark line indicates the median value, and the thick gray lines represent the 25th and 75th percentiles from the distribution made of all median PMTH obtained from the sessions included in the group. Thin gray lines represent the 5th, median, and 95th from the distribution made of all median PMTH obtained from surrogate raster plots from the sessions included in the group. Black asterisk indicate that the median value is above the 95th percentile or below the 5th percentile from the surrogate dataset (**B**). Same as (**A**), but illustration is made with one P12 *Gad1^Cre/+* mouse pup and PMTHs are built with all imaging sessions from pups aged between P10 and P12 (N = 7 mice, n = 9 imaging sessions). Note the presence of red labeled processes in the neuropil of the *stratum pyramidale* of P12 in contrast to P5. Results to build the PMTHs, as well as Calcium Imaging Complete Automated Data Analysis (CICADA) configuration files to reproduce the analysis, are available in *Figure 3—source data 1*.

The online version of this article includes the following source data and figure supplement(s) for figure 3:

**Source data 1.** Analysis configuration files and numerical data used in *Figure 3*.

**Figure supplement 1.** Interneuron identification based on genetic labeling and deep learning classification.

**Figure supplement 2.** Fluorescence changes in pyramidal cells and interneurons focused on the onset of movement.

**Figure supplement 3.** Retrograde tracing from pyramidal cells and interneurons during the first two postnatal weeks.

the synaptic inputs driving both cell types. In the following, we have addressed both, nonmutually exclusive, hypotheses.

We first compared the developmental time course of extra-hippocampal synaptic afferences onto CA1 GABAergic neurons and pyramidal cells using a rabies retrograde tracing method (*Wickersham*

*et al., 2007*). We focused on changes that may occur around the end of the first postnatal week. To do so, two groups were compared, an *early* (AAV1-hSyn-FLEX-nGToG-WPRE3 – helper virus – injected at P0; SAD-B19-RVdG-mCherry – pseudotyped-defective rabies virus – at P5; and immunohistochemistry [IHC] at P9, *Figure 3—figure supplement 3A–C*) and a *late* one (AAV1-hSyn-FLEX-nGToG-WPRE3 – helper virus – injected at P0; SAD-B19-RVdG-mCherry – pseudotyped defective rabies virus – at P9; IHC at P13, *Figure 3—figure supplement 3D–F*). Injections were performed in either *GAD1^{Cre/+}* or *Emx^{Cre/+}* pups in order to specifically target GABAergic or glutamatergic cells, respectively. Four *GAD1^{Cre/+}* pups (two early and two late injections) and three *Emx^{Cre/+}* pups (one early and two late) were analyzed with injection sites restricted to the hippocampus. Starter and retrogradely labeled cells were found all over the ipsilateral hippocampus. For both *GAD1^{Cre/+}* and *Emx^{Cre/+}* pups, we found no striking difference in the retrogradely labeled extra-hippocampal regions between the *early* and *late* groups. In agreement with previous studies (*Supèr and Soriano, 1994*), we found that GABAergic and glutamatergic neurons in the dorsal hippocampus received mainly external inputs from the ento-rhinal cortex, medial septum, and contralateral CA3 area (retrogradely labeled cells in these regions were found in four out of four *GAD1^{Cre/+}* pups and three out of three *Emx^{Cre/+}* pups, *Figure 3—figure supplement 3*). Thus, we could not reveal any major switch in the nature of the extra-hippocampal inputs impinging onto local CA1 neurons. Thus, we next explored the maturation of local somatic GABAergic innervation given its significant evolution throughout that period (*Jiang et al., 2001*; *Marty et al., 2002*; *Morozov and Freund, 2003*) as well as our observation of a dense tdTomato signal in the pyramidal layer from *GAD1^{Cre/+}* mouse pups at P12 (*Figure 3B*), not visible at P5 (*Figure 3A*).

## Abrupt emergence of a functional somatic GABAergic innervation at the beginning of the second postnatal week

We first analyzed the anatomical development of somatic GABAergic innervation within the CA1 pyramidal layer from P3 to P11, focusing on the innervation from putative parvalbumin-expressing basket cells (PVBCs), its main contributor. To this aim, we performed immunohistochemistry against Synaptotagmin2 (Syt2), which has been described as a reliable marker for parvalbumin-positive inhibitory boutons in cortical areas (*Figure 4A*, *Sommeijer and Levelt, 2012*). Using a custom-made Fiji plugin (RINGO, see 'Materials and methods'), we quantified the surface of the pyramidal cell layer covered by Syt2 labeling at different stages and found that between P3 and P7, PV innervation remained stable (median values: P3: 0.34%; P5: 0.57%; P7: 0.49%, three mice per group, *Figure 4B*). However, after P9, an increase in the density of positive labeling was observed (P9: 1.03%; P11: 1.48%, three mice per group, *Figure 4B*). These results are consistent with previous work (*Jiang et al., 2001*; *Marty et al., 2002*), as well as with our tdTomato labeling (*Figure 3A and B*) and GCaMP imaging (*Figure 4—figure supplement 1*, *Video 7*). They also match the transition observed in CA1 dynamics (*Figure 3*). We next tested whether GABAergic axons in the pyramidal layer were active during periods of movement. We restricted the analysis of these experiments to P9–10 as axonal arborization inner-vating the CA1 pyramidal layer was not present before (i.e., there was no fluorescent signal before P9 in the *stratum pyramidale*, *Figure 4—figure supplement 1*, *Video 7*). To do so, we restrained the expression of the calcium indicator GCaMP6s to the axon (*Broussard et al., 2018*) of interneurons using *GAD1^{Cre/+}* mouse pups and specifically imaged axonal arborization in the pyramidal cell layer (*Figure 4C*, left panel). Fluorescence signals were extracted from axonal branches using PyAmnesia (a method to segment axons, see 'Materials and methods,' *Figure 4C*, right panel), and then normalized using DF/F. As expected (see *Figure 3B*), an increase in the fluorescent signal from GABAergic axonal branches was observed following movement (P9–10: n = 3 mice, *Figure 4D*). As a result, we reasoned that the emergence of functional perisomatic GABAergic activity could contribute to the reduction in activity observed after movement during the second postnatal week in pyramidal neurons.

## Increasing feedback inhibition in two-population models explains the developmental transition

To test whether an increase in perisomatic inhibition alone can explain the switch in network dynamics between the first and second postnatal weeks, we simulated a two-population network model mimicking the development of perisomatic innervation (*Figure 5A*, see 'Materials and methods'). Using a rate model and a leaky integrate and fire (LIF) model, we show that increasing the number of perisomatic inhibitory connections can account for the experimentally observed decrease in responses

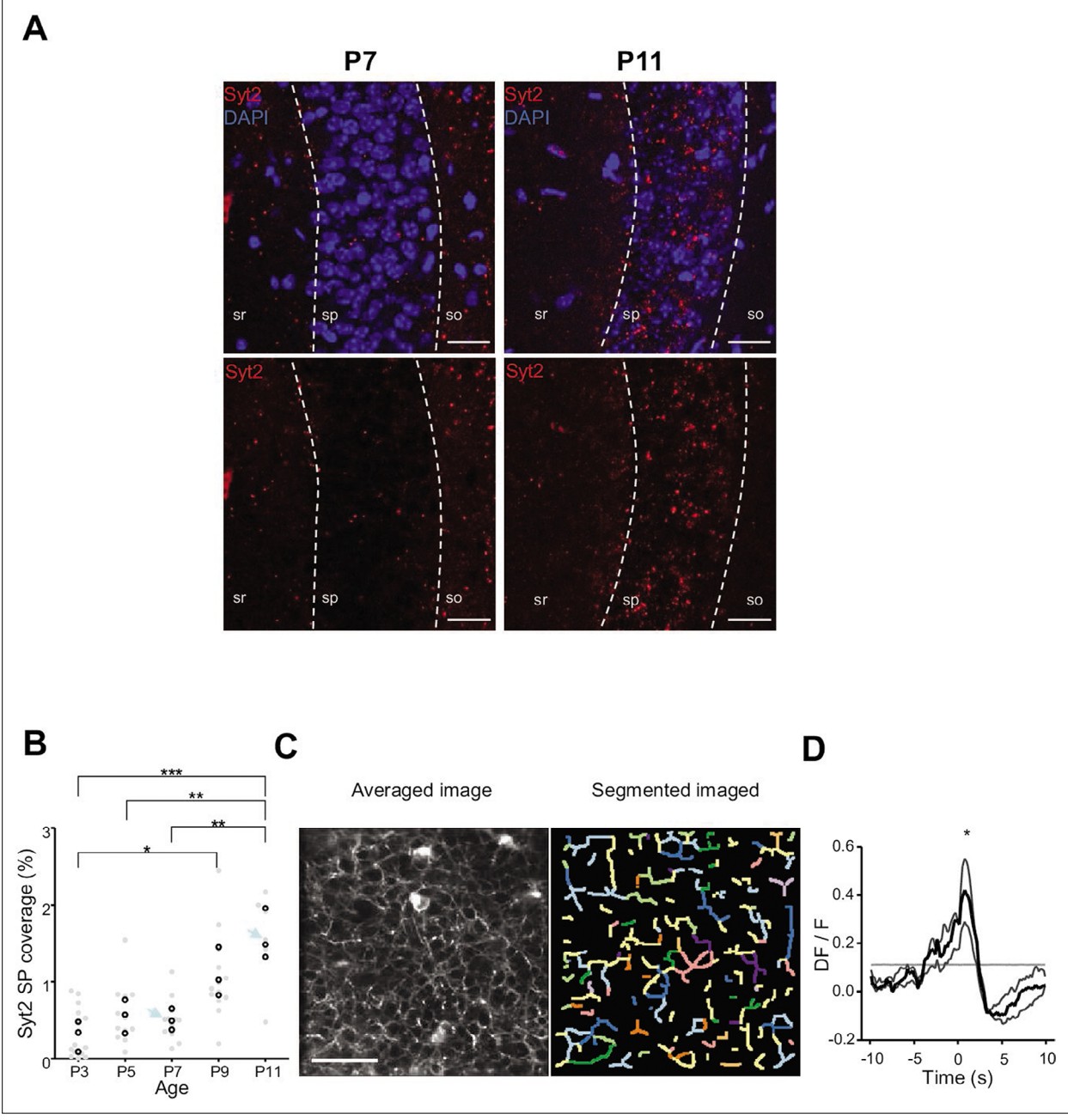

**Figure 4.** Emergence of perisomatic GABAergic innervation. (**A**) Representative example confocal images of the CA1 region in a P7 (left) and P11 (right) mouse pup. DAPI staining was used to delineate the *stratum pyramidale* (sp) from the *stratum radiatum* (sr) and *stratum oriens* (so, top row). Synaptotagmin-2 labeling (Syt2) is shown in the top and bottom rows. Illustrated examples are indicated by red dots in the associated quantification in (**B**). (**A**) Scale bar = 50 µm. (**B**) Fraction of the pyramidal cell layer covered by Syt2-positive labeling as a function of age. Each gray dot represents the average percentage of coverage from two images taken in the CA1 region of a hippocampal slice. Open black dots are the average values across brain slices from one mouse pup. Blue arrows indicate the slices used for illustration in (**A**). A significant effect of age was detected (one-way ANOVA, $F$ = 13.11, p=0.0005, three mice per age group). Multiple-comparison test shows a significant difference between age groups (Bonferroni's test, *p<0.05, **p<0.01, ***p<0.001). (**C**) Averaged image of a field of view in the pyramidal cell layer of a P9 *Gad1*$^{Cre/+}$ mouse pup injected with a Cre-dependent Axon-GCaMP6s indicator (left) and the segmented image resulting from PyAmnesia (right). (**C**) Scale bar = 50 µm. (**D**) Peri-movement time histogram (PMTH) showing the DF/F signal centered on the onsets of animal movement (N = 3 mice, n = 3 imaging sessions). The dark gray line indicates the median value from the surrogate. Results obtained from surrogates are represented by light gray lines. Black asterisk indicate that the median value is above the median from the surrogate.

The online version of this article includes the following source data and figure supplement(s) for figure 4:

**Source data 1.** Analysis configuration files and numerical data used in *Figure 4*.

*Figure 4 continued on next page*

*Figure 4 continued*

**Figure supplement 1.** In vivo visualisation of GABAergic axonal arborisation during the first two postnatal weeks in the region CA1 of the hippocampus.

to movement-like feedforward inputs (*Figure 5B*). Time constants of the rate model and synaptic time constants of the LIF model were chosen to match the slow kinetics of synaptic transmission that exist at early developmental stages (see *Figure 5—source data 1*). Faster excitatory and inhibitory timescales, on the order of a few milliseconds, generate network dynamics that could not be followed by our calcium sensor. We chose to model them by simply providing a noisy, normally distributed, input to all the cells. Durations of feedforward inputs were chosen similar to experimental movement durations (see *Figure 5—figure supplement 1A* for a log-normal fit of the movement durations). When inhibition is weak, the average activity of the pyramidal neurons increases at the onset of a given twitch. Then, it quickly relaxes to the baseline with a timescale that follows the synaptic time constant (*Figure 5B*, left panel). In the presence of strong inhibition, there is a reduction in response to movement inputs. In addition, due to strong feedback inhibition following the movement responses, network activity relaxes to the baseline with an undershoot (*Figure 5B*, right panel), recapitulating the experimental findings (see *Figure 2A*). Similarly, PMTHs for interneurons were obtained (see *Figure 5—figure supplement 1B*).

The inhibition of principal cells following movements after P9, revealed by the PMTHs, could result from the development of a direct external inhibitory input or from changes in local circuits. In order to further examine the nature of the interaction between local pyramidal cells and interneurons displayed a direct interaction in the absence of the external movement-related input, we next used our experimental data to compute the cross-correlograms between interneurons and pyramidal cells calcium transients (inferred by DeepCINAC [*Denis et al., 2020*]) in the periods without movement (P5–8 light gray curve, P9–12 dark curve, *Figure 5C*, left panel). For P9–12, we observed a rapid drop in the correlation at positive time points, suggesting a feedforward inhibition of principal cells' activity. Such drop was absent for P5–8, where the cross-correlogram was symmetric and centered at zero. Both the rate and LIF models displayed similar activity correlograms as the experimental data, with an undershoot in the presence of strong inhibitory feedback for positive time points (*Figure 5C*, middle and right panels). Thus, increasing $J_{EI}$ has the effect of strengthening the cross-correlation undershoot. $J_{EI}$ for both models were chosen to match the amplitude of this undershoot while at the same time matching the inhibition observed in the PMTHs. Auto-correlograms of the excitatory and inhibitory activity were also measured and compared to our model predictions (*Figure 5—figure supplement 1C*).

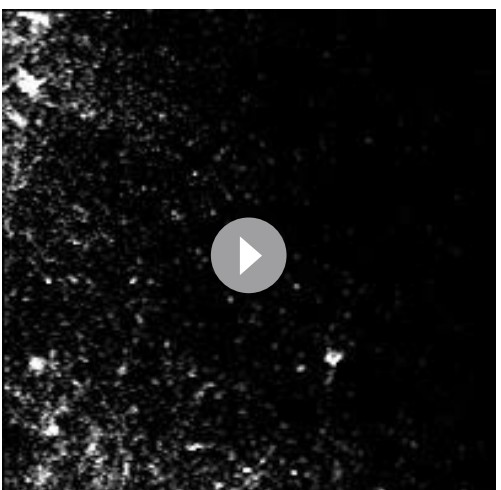

**Video 7.** Calcium imaging movie from the field of view (FOV) shown in the middle panel of Figure 4—figure supplement 1B.
https://elifesciences.org/articles/78116/figures#video7

The consistency of our models with the experimental cross-correlograms, which were computed from the activity recorded during periods of immobility, further shows that the observed network dynamics and, in particular, the correlation undershoots most likely result from recurrent perisomatic inhibition rather than a feedforward drive from upstream areas. Therefore, in our model, the maturation of perisomatic inhibition alone was sufficient to support a switch in network dynamics.

## Discussion

Using for the first time in vivo two-photon calcium imaging in the hippocampus of non-anesthetized mouse pups and a deep-learning-based approach to infer the activity of principal cells and interneurons, we show that the end of the first postnatal week marks a salient step in the anatomical and functional development of the CA1 region.

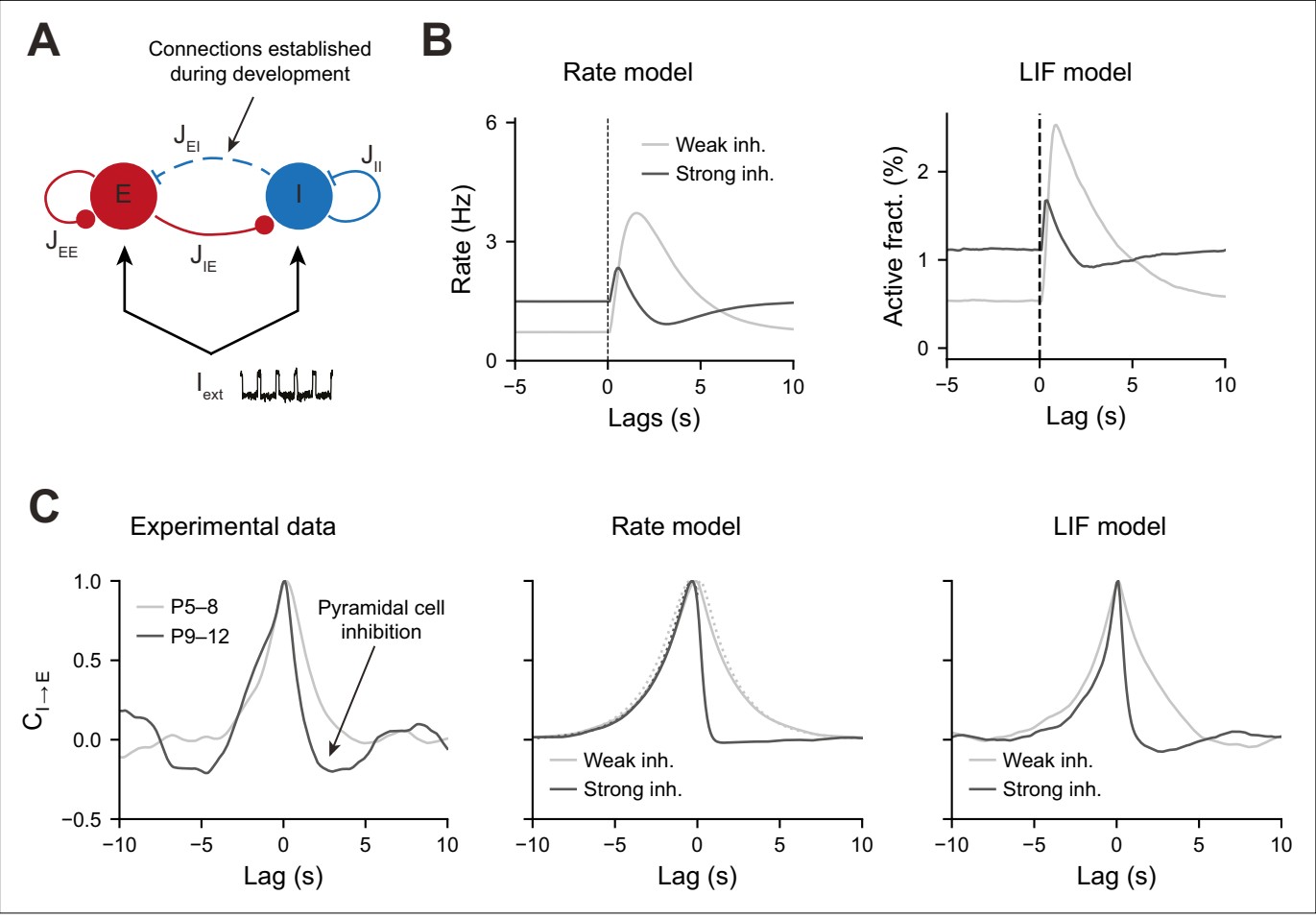

**Figure 5.** Modeling the effects of perisomatic inhibition on pyramidal cell response. (**A**) The model consists of two populations, Excitatory (E) and Inhibitory (I) receiving feedforward input $I_{ext}$. The interaction strengths, Jab, represent the effect of the activity of population b on a. We studied the effects of perisomatic inhibition on the activity of pyramidal cells by varying the parameter $J_{EI}$ (i.e., the number of I to E connections). For the rate model, the overall scale of the rates is arbitrary. For the leaky integrate and fire (LIF) model, the parameters were tuned to match the percentages of active cells per experimental bin (100 ms). (**B**) Peri-movement time histogram (PMTH) response of excitatory neurons to pulse input in the rate model and LIF network. (**C**) Cross-correlations during periods of immobility, in experimental data (left), rate model (middle), and LIF network (right). In the rate model, dotted lines are the predicted correlation from the analytic expressions and solid lines are the results from numerical integration. In all the simulations, the signals were convolved with an exponential kernel of characteristic time 2 s to account for GCamp6s decay time.

The online version of this article includes the following source data and figure supplement(s) for figure 5:

**Source data 1.** Table of the model parameter values.

**Figure supplement 1.** Modeling the effects of perisomatic inhibition on CA1 correlated activity.

Indeed, within 2 days (P8–10), the link between CA1 principal cells' activity and self-triggered movements is inverted and neurons are preferentially active during immobility periods. This is likely due to the time-locked anatomical and functional rise of somatic GABAergic activity, given that interneurons remain highly active throughout this period, including in response to spontaneous movements. In this way, CA1 circuits start detaching from external inputs. Given the importance of local dynamics for hippocampal function and cortical circuits operation in general, this is likely to be a critical general step in the proper maturation of cognitive circuits.

## Early postnatal calcium activity in CA1 is driven by sensorimotor inputs

We found that, until P7–9, spontaneous movements are followed by a significant peak in calcium events in the CA1 principal cell layer and that most neuronal activity occurs during SCEs. This early link between sensorimotor inputs and early cortical dynamics has been previously reported using electrophysiological recordings in various areas and species, including humans (**Milh et al., 2006**). Here,

we extend that observation to calcium transients, which not only indirectly report action-potential firing as well as other modes of cell activation during development but also critically regulate activity-dependent genetic processes. In addition, we could describe the response to these movements with single-cell resolution. Of note, it is important to keep in mind that part of the overlying cortex, including the primary sensory cortex, was removed to grant optical access to the hippocampus. This region may contribute to relaying the sensory feedback from the twitches to the hippocampus (*Khazipov and Milh, 2018*; *Valeeva et al., 2019a*). The surgical procedure may thus damage incoming axons from the temporoammonic track linking the entorhinal cortex to the hippocampus. Accordingly, we observed (1) that the CA1 response to movements from the contralateral limbs was slightly reduced, (2) the eSW frequency in the ipsilateral hemisphere was slightly diminished, and (3) there was a small increase in the power spectra of network oscillations below 20 Hz. In addition, it cannot be excluded that other movements that we have not detected, like whisker movements, could also contribute to the patterning of CA1 activity. It is also possible that self-generated activity from other sensory organs but independent from movement, like the retina or the olfactory bulb, also contributes to hippocampal dynamics. Interestingly, in contrast to previous reports (*Tiriac and Blumberg, 2016*), we could not observe any significant difference between twitches (occurring mainly during active sleep) and longer, more complex movements (associated with wakefulness). In one mouse pup, we directly combined two-photon imaging with EMG recordings to better define brain states and confirmed that both REM and wake-associated movements were followed by an activation of CA1 neurons, with the latter triggering a smaller response. This difference from previous reports (*Tiriac and Blumberg, 2016*) may reveal a difference between calcium imaging and electrophysiology, the former sampling from a larger population but at a lower temporal and spike signal resolution. The patterning of CA1 dynamics in the large imaged population did not reveal any obvious spatial distribution for movement-activated cells but we cannot exclude that these would vary along the radial and transverse directions, which are the two main axes of principal cell development (*Caviness, 1973*), and are differentially targeted by perisomatic PVBCs (*Lee et al., 2014*; *Valero et al., 2015*).

Passed the end of the first postnatal week, between P8 and P10, a significant decrease in the fraction of coactive principal cells following movement was observed (while interneurons remained mostly activated by movement). We cannot exclude that some spikes fell below the threshold for the detection of calcium events. In this case, rather than full inhibition, it may be that a strong shortening of the time window for neuronal integration occurred (due to feedback inhibition), which would limit the number of spikes produced by principal cells and thus keep them below detection levels. Yet, a novel machine learning-based algorithm (*Denis et al., 2020*) was used since it was especially designed to infer activity in the dense CA1 pyramidal cell layer. This change in the polarity of principal cells' response to movements is quite abrupt as it happens within less than 2 days (between P8 and 10). This contrasts with the progressive evolution of single-cell-firing frequencies but matches the fast redistribution of neuronal firing towards immobility periods. In this way, hippocampal neuronal dynamics 'internalize' as they stop being driven by movements and preferentially occur within rest. Of note, a small increase in the fraction of active cells can be observed approximately 1 s before the onset of movement in P12 pups, indicating that activity would start building up in CA1 prior to movement. A corollary discharge would increase activity prior to movement on a much shorter timescale. There is therefore no obvious explanation for this interesting phenomenon. Anticipatory cell firing prior to locomotion has been previously reported in the adult cortex (e.g., see *Vinck et al., 2015*). Different mechanisms could support such anticipatory firing, including the influence of top-down inputs, changes in arousal states, or any complex neuromodulatory interactions possibly associated with changes in the sleep–wake cycle and that could involve, for example, the norepinephrine, serotonin, or acetylcholine systems.

This 'internalization' of hippocampal dynamics is reminiscent of similar phenomena observed in other cortical areas, such as the barrel cortex where whisker stimulation induces a reduction in the size of cell assemblies following P9 while the same stimulation widens cell assembly size a few days before (*Modol et al., 2019*). It is also reminiscent of the recently described transient quiescent period observed in the somatosensory cortex using extracellular electrophysiological recordings (*Domínguez et al., 2021*). Lastly, it goes in hand with a sparsification of activity, which is a general developmental process supported by the emergence of inhibition (*Golshani et al., 2009*; *Rochefort et al., 2009*; *Wolfe et al., 2010*).

## Circuit basis for the movement-triggered inhibition of CA1 dynamics

Our results demonstrate that the change in CA1 dynamics occurring at the end of the first postnatal week most likely relies on structural changes in local CA1 circuits rather than rewiring of the long-range extra-hippocampal connectivity.

The long-range circuits mediating the bottom-up flow of self-triggered or externally generated sensory information to the hippocampus are starting to be elucidated. The two main structures directly transmitting sensorimotor information to the dorsal CA1 are the entorhinal cortex and septum. The former processes multisensory information from all sensory cortices (visual, auditory, olfactory, somatosensory), including movement-related sensory feedback (*Rio-Bermudez and Blumberg, 2022*), and was shown to be activated by spontaneous twitches prior to CA1 (*Mohns and Blumberg, 2010*; *Del Rio-Bermudez et al., 2020*; *Valeeva et al., 2019a*) while the latter is more likely to be involved in transmitting internal information (*Fuhrmann et al., 2015*; *Wang et al., 2015*), as well as unexpected environmental stimuli (*Zhang et al., 2018*). In addition to these two canonical pathways, one cannot exclude the involvement of a direct connection from the brainstem, given their existence in the adult and their role in promoting sleep as well as motor twitches (*Liu et al., 2017*; *Szőnyi et al., 2019*). However, our retrograde-tracing experiments did not reveal any direct connection between the CA1 cells and the brainstem at the early ages analyzed here. In addition, we found that both CA1 interneurons and principal cells receive inputs from the septum and entorhinal cortex before the time of the switch (i.e. P9) and that there was no major qualitative change of inputs after, as expected from previous work (*Supèr and Soriano, 1994*). Still, these experiments do not allow a quantitative assessment of the number of inputs nor the type of inputs (GABAergic, cholinergic, etc.), and we cannot fully exclude that a stronger or different source of excitatory drive would be impinging onto interneurons after the switch. Neither can we exclude a functional maturation of those extrinsic inputs. Therefore, future optogenetic and slice physiology work is needed to characterize the bottom-up information flow onto specific components of the local CA1 circuits. Similarly, one cannot exclude a change in the CA3 to CA1 connectivity. Indeed, Schaffer collaterals are known to reach CA1 roughly around the end of the first postnatal week (*Durand et al., 1996*). In addition, roughly at the time of the switch, we do see the emergence of SWRs (*Buhl and Buzsáki, 2005*), a pattern strongly relying on CA3 inputs and perisomatic GABAergic transmission. However, we could not restrict the pool of starter cells to the CA1 region in our retrograde viral-tracing experiments, which precluded analysis of the development of CA3–CA1 connectivity. Interestingly, among the external inputs onto CA1 described above, the entorhinal cortex and CA3 were both shown to exert a mild influence on the organization of intrinsic CA1 dynamics, possibly pointing at a critical role of local interneurons in this process (*Zutshi et al., 2021*).

As indicated by our computational model, the disengagement from movement of CA1 dynamics can be fully explained by the observed rise in anatomical (Syt2 labeling) and functional (axonal GCaMP imaging) connectivity from perisomatic GABAergic cells onto pyramidal cells at the onset of the second postnatal week. This increased connectivity could not be easily captured with our retrograde viral labeling since the absence of early PV expression precludes the identification of PVBCs, the most prominent subtype of perisomatic GABAergic cells, among retrogradely labeled cells in *Emx*$^{Cre/+}$ pups. Early anatomical studies had already indicated that an increase in somatic GABAergic inhibition, including from CCK-basket cells, occurred in CA1 during the first postnatal week (*Gour et al., 2021*; *Danglot et al., 2006*; *Jiang et al., 2001*; *Marty et al., 2002*; *Morozov and Freund, 2003*). However, this rise was expected to be more progressive and not as abrupt as observed here as it happened within 2 days. If the axonal coverage of the *stratum pyramidale* by PV-basket cells axons increases, we cannot exclude that this is a general phenomenon, concerning all perisomatic subtypes, including soma-targeting CCK-expressing basket cells that develop anatomically at around the same time (*Morozov and Freund, 2003*) or chandelier cells. In addition, our computational model indicates that the emergence of feedback inhibition is sufficient to reproduce the developmental shift observed here, which could also involve other types of CA1 interneurons, including dendrite-targeting ones.

Interestingly, a similar rise of somatic GABAergic axonal coverage occurs in the barrel cortex at the same time. Indeed, recent connectomic mapping using 3D electron microscopy in that region revealed that the preferential targeting of cell bodies by GABAergic synapses increased almost three-fold between postnatal days 7 and 9 (*Gour et al., 2021*), whereas two-photon imaging of putative GABAergic somatic axons in the same region revealed broader domains of co-activation (*Modol*

*et al., 2019*). This time period for the shift may be synchronous within brain regions involved in sensorimotor integration such as the hippocampus and somatosensory cortex. Otherwise, PV expression was shown to develop sequentially in a region-specific manner *Reh et al., 2020* following their intrinsic developmental age (*Donato et al., 2017*).

We found that many principal cells are inhibited by movement while most imaged GABAergic cells remained activated during the second postnatal week. This therefore indirectly suggests a net inhibitory effect of GABAergic transmission after the first postnatal week. This is expected since the shift from excitatory to inhibitory synaptic transmission was reported to occur earlier in the hippocampus in vivo (*Murata and Colonnese, 2020*). On a side note, the lack of somatic GABAergic inputs before P7 indicates that the early excitatory GABAergic drive in CA1 circuits likely originates from nonsomatic GABAergic interneurons, which include long-range, dendrite-targeting or interneuron-specific interneurons. The circuit role of excitatory GABAergic transmission should be revisited taking into account this new finding.

The movement-associated inhibition could result equally from feedforward (direct activation of interneurons from movement-transmitting inputs such as the entorhinal cortex) or feedback (from local CA1 cells) inhibition. Our analysis of pairwise correlations in the absence of movement, as well as our computational simulations, indicates the latter. A similar strengthening of feedback inhibition has previously been observed in the developing somatosensory cortex (*Anastasiades and Butt, 2012*). The inhibition of activity following movement is likely to be occurring during a transient developmental period. Indeed, in the adult, both interneurons and principal cells usually increase their activity as the animal moves (*Fuhrmann et al., 2015*). Therefore, the switch observed here opens another developmental time window that probably closes with the emergence of perineuronal nets and cell activation sequences at the end of the third developmental week (*Farooq and Dragoi, 2019*; *Horii-Hayashi et al., 2015*; *Muessig et al., 2019*). We would like to propose this developmental window to be the critical period for CA1 development, a period during which experience-dependent plasticity can be observed.

## Conclusion

Cognitive hippocampal maps rely on two forms of representation, one that is map-based or allocentric and the other that is self-referenced, or egocentric and requires body movement. We would like to propose that the early postnatal period described here, where the hippocampus learns the statistics of the body, and which terminates with the rise of a recurrent inhibitory network, is a key step for the emergence of self-referenced representations onto which exploration of the external world can be grafted. An imbalance between self-referenced and environmental hippocampal representations due to a miswiring of local somatic inhibition could have major outcomes. It could be on the basis of several neurodevelopmental disorders, including autism spectrum disorders (ASDs) and schizophrenia. Interestingly, both disorders have been associated with an aberrant maturation of PV-expressing interneurons (*Gogolla et al., 2014*; *Jurgensen and Castillo, 2015*; *Lewis et al., 2005*). In addition, the proper development of the peripheral sensory system, which is partly initiating the early CA1 dynamics reported in our study, is also critically involved in ASD (*Orefice et al., 2016*). The period described here corresponds to the third trimester of gestation and likely extends postnatally given the protracted integration of GABAergic interneurons into functional circuits in the human brain (*Murphy et al., 2005*; *Paredes et al., 2016*). Future work should determine when a similar rise in somatic inhibition occurs in human infants and test whether it could constitute a valuable biomarker for cognitive neurodevelopmental disorders.

## Materials and methods

### Mice

All experiments were performed under the guidelines of the French National Ethics Committee for Sciences and Health report on 'Ethical Principles for Animal Experimentation' in agreement with the European Community Directive 86/609/EEC (Apafis #18-185 and #30-959).

### Experimental procedures and data acquisition

#### Viruses

In vivo calcium imaging experiments were performed using AAV1-hSyn-GCaMP6s.WPRE.SV40 (pAAV.Syn.GCaMP6s.WPRE.SV40 was a gift from Douglas Kim & GENIE Project [Addgene viral

prep# 100843-AAV1; http://n2t.net/addgene:100843; RRID:Addgene_100843]), AAV9-FLEX-CAG-tdTomato (pAAV-FLEX-tdTomato was a gift from Edward Boyden [Addgene viral prep# 28306-AAV9; http://n2t.net/addgene:28306; RRID:Addgene_28306]), AAV9-hSyn-FLEX-axon-GCaMP6s (pAAV-hSynapsin1-FLEx-axon-GCaMP6s was a gift from Lin Tian [Addgene viral prep# 112010-AAV9; http://n2t.net/addgene:112010; RRID:Addgene_112010]). Retrograde-tracing experiments were performed using AAV1-hSyn-FLEX-nGToG-WPRE3 (Charité# BA-096) and SAD-B19-RVdG-mCherry (gift from the Conzelmann laboratory).

## Intracerebroventricular injection

This injection protocol was adapted from already published methods (*Rübel et al., 2021Kim et al., 2014*). Mouse pups were anesthetized on ice for 3–4 min, and 2 µL of viral solution (titration at least $1 \times 10^{13}$ vg/mL) was injected into the left lateral ventricle whose coordinates were estimated at the 2/5 of the imaginary line between the lambda and the eye at a depth of 0.4 mm. Correct injection was visualized by the spreading of the viral-dye mixture (1/20 of fast blue). In SWISS mouse pups. we injected 2 µL of AAV2.1-hSyn-GCAMP6s.WPRE.SV40; in *GAD1^{Cre/+}* mouse pups we injected either a mix of 1.3 µL of AAV2.1-hSyn-GCAMP6s.WPRE.SV40 with 0.7 µL of AAV9-FLEX-CAG-tdTomato or 2 µL of AAV9-hSyn-FLEX-axon-GCaMP6s.

## Intra-hippocampal injection

When hippocampal viral injections were performed at P0 (AAV1-hSyn-FLEX-nGToG-WPRE3), mouse pups were anesthetized by inducing hypothermia on ice and maintained on a dry ice-cooled stereotaxic adaptor (Stoelting, #51615) with a digital display console (Kopf, model 940). Dorsal hippocampus was targeted by empirically determined coordinates, based on the Atlas of the Developing Mouse Brain (*Paxinos and Watson, 2020*), using transverse sinus and superior sagittal sinus as reference: 0.8 mm anterior from the sinus intersection; 1.5 mm lateral from the sagittal sinus; 1.1 mm depth from the skull surface. Under aseptic conditions, an incision was made in the skin, the skull was exposed, and gently drilled (Ball Mill, Carbide, #¼ 0.019″ –0.500 mm diameter, CircuitMedic). Then, 10 nL of undiluted viral solution was injected using an oil-based pressure injection system (Nanoject III, Drummond Scientific, rate of 5 nL/min). The tip of the pipette was broken to achieve an opening with an internal diameter of 30–40 µm. When hippocampal viral injections were performed at P5 or P9 (SAD-B19-RVdG-mCherry), pups were anesthetized using 3% isoflurane in a mix of 90% $O_2$–10 % air and maintained during the whole surgery (~0:30 hr) between 1 and 2.5% isoflurane. Body temperature was monitored and maintained at 36°C. Analgesia was controlled using buprenorphine (0.05 mg/kg). Under aseptic conditions, an incision was made in the skin, the skull was exposed, and anteroposterior and mediolateral coordinates of the dorsal hippocampus were estimated by eye looking at the skull sutures. The skull was gently drilled and 10 nL of a viral solution was injected (Nanoject III, Drummond Scientific, rate of 5 nL/min) at a depth of 1.25 mm below the dura.

## Window implant surgery

The surgery to implant a 3-mm-large cranial window above corpus callosum was adapted from previous methods (*Villette et al., 2015*). Anesthesia was induced using 3% isoflurane in a mix of 90% $O_2$–10% air and maintained during the whole surgery (~1:30 hr) between 1 and 2.5% isoflurane. Body temperature was controlled and maintained at 36°C. Analgesia was controlled using buprenorphine (0.05 mg/kg). Coordinates of the window implant were visually estimated. Then a small custom-made headplate was affixed with cyanoacrylate and dental acrylic cement. The skull was removed and the cortex was gently aspirated until the appearance of the external capsule/alveus. At the end of the cortectomy, we sealed a 3 mm glass window diameter circular cover glass (#1 thickness, Warner Instrument) attached to a 3-mm-diameter large and 1.2-mm-height cannula (Microgroup INC) with Kwik-Sil adhesive (WPI) and fixed the edge of the glass with cyanoacrylate. We let the animal recover on a heated pad for at least 1 hr before the imaging experiment.

## Imaging

Two-photon calcium imaging experiments were performed on the day of the window implant using a single-beam multiphoton pulsed laser scanning system coupled to a microscope (TriM Scope II, LaVision Biotech). The Ti:sapphire excitation laser (Chameleon Ultra II, Coherent) was operated at

920 nm. GCaMP fluorescence was isolated using a bandpass filter (510/25). Images were acquired through a GaAsP PMT (H7422-40, Hamamatsu) using a ×16 immersion objective (NIKON, NA 0.8). Using Imspector software (LaVision Biotech), the fluorescence activity from a 400 μm × 400 μm field of view was acquired at approximately 9 Hz with a 1.85 μs dwell time per pixel (2 μm/pixel). Imaging fields were selected to sample the dorsal CA1 area and maximize the number of imaged neurons in the *stratum pyramidale*. Piezo signal, camera exposure time, and image triggers were synchronously acquired and digitized using a 1440A Digidata (Axon Instrument, 50 kHz sampling) and the AxoScope 10 software (Axon Instrument). During the imaging session, body temperature is continuously controlled.

## Behavioral recordings

Simultaneously with imaging experiments, mouse motor behavior was monitored. In a first group of animals, motor behavior was monitored using two or three piezos attached to the paws of the animal. The signal from the piezo was acquired and digitized using a 1440A Digidata and the AxoScope 10 software. In a second group of animals, pups were placed and secured on an elevated platform (with the limbs hanging down on each side without touching the ground nor the support, as described here; *Blumberg et al., 2015*). Motor behavior was monitored using two infrared cameras (Basler, acA1920-155um) positioned on each side of the animal. For each camera, a square signal corresponding to the exposure time of each frame from the camera was acquired and digitized using a 1440A Digidata and the AxoScope 10 software. If the number of behavior frames from the square signal was higher than the number of saved frames (meaning that some camera frames were dropped during the acquisition), the imaging session was excluded from any movement related analysis (see *Supplementary file 1*).

## Recording of EMG activity in neonatal mice

The vigilance state of neonatal mice was assessed through analysis of EMG signals obtained from a single insulated tungsten wire (A-M Systems 795500) implanted in the nuchal muscle. A stainless steel wire (A-M Systems 786000) wire inserted on the skull surface above the cerebellum and secured in place with dental cement served as the reference electrode. Signals from the electrodes were first passed through a headstage pre-amplifier before being digitized at 16,000 Hz (Digital Lynx SX, Neuralynx [the pre-amplifier and digitizer were both from Neuralynx, as was the acquisition software, Cheetah]) and saved to a hard disk. TTL signals from the imaging and camera acquisition systems were simultaneously recorded as well to enable precise synchronization of EMG recordings with the camera and imaging data.

## In vivo extracellular electrophysiological recordings

Multisite probes (16-channel silicon probes with 50 μm separation distance, NeuroNexus, USA) were used to record electrophysiological activity below the window implant and in the intact hippocampus. To do so, we positioned the mouse pup (that had previously undergone a window implant) on the experimental setup. To head-fix the animal, the skull surface was covered with a layer of dental acrylic except the area above the intact hippocampus. In the intact (contralateral) hippocampus, the electrodes were positioned using the stereotaxic coordinates of approximately 1.5 mm anterior to lambda and 1.5 mm lateral from the midline. Hippocampus under the window was recorded through the hole drilled in the window implant. Both multisite silicon probes were positioned at the depth to record *strata oriens* (SO), *pyramidale* (SP), *radiatum* (SR), and *lacunosum moleculare* (SLM). After the positioning of the electrodes, the animal was left in the setup for 1 hr to recover followed by 2 hr recordings of the neuronal activity in both hippocampi simultaneously.

## Histological processing

Pups were deeply anesthetized with a mix of Domitor and Zoletil (0.9 and 60 mg/kg, respectively), then transcardially perfused with 4% paraformaldehyde (PFA) in 0.1 M phosphate-buffered saline (PBS) (PBS tablets, 18912-014, Life Technologies). For perisomatic innervation analysis, brains were post-fixed overnight at 4°C in 4% PFA in 0.1 M PBS, washed in PBS, cryo-protected in 30% sucrose in PBS, before liquid nitrogen freezing. Brains were then sectioned using a cryostat (CM 3050S, Leica) into 50-μm-thick slices collected on slides. Sections were stored at –20°C until further usage. For tracing experiments, brains from *GAD1*[Cre/+] and *Emx*[Cre/+] pups (The Jackson Laboratory JAX:005628), were

post-fixed overnight at 4°C in 4% PFA in 0.1 M PBS, washed in PBS, and sectioned using a vibratome (VT 1200s, Leica) into sagittal 70–80-µm-thick slices. Sections were stored in 0.1 M PBS containing 0.05% sodium azide until further usage. Immunocytochemistry was processed as described previously (*Bocchio et al., 2020*). Briefly, sections were blocked with PBS-Triton (PBST) 0.3 and 10% normal donkey serum (NDS), and incubated with a mix of up to three primary antibodies simultaneously diluted in PBST with 1% NDS overnight at room temperature with the following primary antibodies: rabbit anti-dsRed (1:1000; Clontech, AB_10013483), chicken anti-GFP (1:1000, Aves Labs, GFP-1020, AB_10000240), and mouse anti-synaptotagmin-2 (1:100; Developmental Studies Hybridoma Bank, AB_2315626). After several washes, according to the mixture of primary antibodies, the following secondary antibodies were used: donkey anti-chicken Alexa 488 (1:500, SA1-72000), donkey anti-rabbit Alexa 555 (1:500, Thermo Fisher, A31570), donkey anti-mouse Alexa 488 (1:500, Thermo Fisher, A21202), and donkey anti-mouse Alexa 647 (1:500, Thermo Fisher, A31571). After Hoechst counter-staining, slices were mounted in Fluoromount. Epifluorescence images were obtained with a Zeiss AxioImager Z2 microscope coupled to a camera (Zeiss AxioCam MR3) with an HBO lamp associated with 470/40, 525/50, 545/25, and 605/70 filter cubes. Confocal images were acquired with a Zeiss LSM-800 system equipped with a tunable laser providing excitation range from 405 to 670 nm. For quantifying synaptotagmin-2, 11-µm-thick stacks were taken (z = 1 µm, pixel size = 0,156 µm) with the confocal microscope using a Plan-Achromat ×40/1.4 oil DIC objective.

## Data preprocessing

### Motion correction

Image series were motion corrected either by finding the center of mass of the correlations across frames relative to a set of reference frames (*Miri et al., 2011*) or using the NoRMCorre algorithm available in the CaImAn toolbox (*Pnevmatikakis and Giovannucci, 2017*), or both.

### Cell segmentation

Cell segmentation was achieved using Suite2p (*Pachitariu et al., 2017*). Neurons with pixel masks including processes (often the case for interneurons located in the *stratum oriens*) were replaced by soma ROI manually drawn in ImageJ and matched onto Suite2p contours map using Calcium Imaging Complete Automated Data Analysis (CICADA; source code available on Cossart lab GitLab group ID: 5948056). In experiments performed on $GAD1^{Cre/+}$ animals, tdTomato-labeled interneurons were manually selected in ImageJ and either matched onto Suite2p contours map or added to the mask list using CICADA.

### Axon segmentation

Axon segmentation was performed using pyAMNESIA (a *P*ython pipeline for analyzing the *A*ctivity and *M*orphology of *NE*urons using *S*keletonization and other *I*mage *A*nalysis techniques; source code available on Cossart lab GitLab group ID: 5948056). pyAMNESIA proposes a novel image processing method based on three consecutive steps: (1) extracting the axonal morphology of the image (skeletonization), (2) discarding the detected morphological entities that are not functional ones (branch validation), and (3) grouping together branches with highly correlated activity (branch clustering). To extract the skeleton, we first performed 3D Gaussian smoothing and averaging of the recording, producing an image that summarizes it; on this image are then successively applied a histogram equalization, a Gaussian smoothing, an adaptive thresholding, and finally a Lee skeletonization (*Suen et al., 1994*), allowing for the extraction of the skeleton mask and the morphological branches. To ensure the functional unity of the segmented branches, we only kept those that illuminated uniformly, where uniformity was quantified by the skewness of the pixel distribution of the branch during a calcium transient (branch validation). To cluster the valid branches based on their activity, we first extracted their average trace – being the average image intensity along the branch for each frame – and then clustered the branches traces using t-SNE and HDBSCAN algorithms with Spearman's correlation metric (branch clustering).

### Cell-type prediction

Cell-type prediction was done using the DeepCINAC cell-type classifier (*Denis et al., 2020*). We used imaging sessions from $GAD1^{Cre/+}$ mouse line injected with a viral mixture of AAV2.1-hSyn-GCAMP6s.

WPRE.SV40 and AAV9-FLEX-CAG-tdTomato allowing us to manually identify in our recordings genetically labeled interneurons to train and test a cell-type classifier. Overall, we used 643 cells (245 labeled interneurons, 245 putative pyramidal cells, and 153 noisy cells) to train the cell-type classifier and 100 cells (38 labeled interneurons, 51 putative pyramidal cells, and 11 noisy cells) to evaluate its performance. Briefly, a neuronal network composed of a convolutional neural network (CNN) and long short-term memory (LSTM) was trained using labeled interneurons, pyramidal cells, and noisy cells to predict the cell type using 100-frame-long movie patches centered on the cell of interest. Each cell was classified as interneuron, pyramidal cell, or noise. Cells classified as 'noisy cells' were removed from further analysis. 'Labeled interneurons' were first kept in a separate cell-type category and added to the interneurons list that were inferred with a 91% precision.

### Activity inference

Activity inference was done using DeepCINAC classifiers (*Denis et al., 2020*). Briefly, a classifier composed of CNN and LSTM was trained using manually labeled movie patches to predict neuronal activation based on movie visual inspection. Depending on the inferred cell type, activity inference was done using either a general classifier or an interneuron-specific classifier. Activity inference resulted in a (cells × frames) matrix giving the probability for a cell to be active at any single frame. We used a 0.5 threshold in this probability matrix to obtain a binary activity matrix considering a neuron as active from the onset to the peak of a calcium transient.

### Behavior

Piezo signals were manually analyzed in a custom-made graphical user interface (Python Tkinter) to label the onset and offset of 'twitches,' 'complex movements,' and 'unclassified movements.' Twitches were defined as brief movements (a few hundred milliseconds long) occurring within periods of rest and detected as rapid deflections of the piezo signal. 'Complex movements' were defined as periods of movement lasting at least 2 s. A few other detected movements could not be categorized based on their duration and occurrence as twitches or complex movements. These are referred to as 'unclassified movements.' These unclassified movements were excluded from the analysis when it is specified that complex movements or twitches only were used but included in analysis when all kinds of movements were combined. Analysis of video tracking was done using CICADA, and behavior was manually annotated in the BADASS (Behavioral Analysis Data And Some Surprises) GUI. If camera frames were dropped during the acquisition, the imaging session was excluded from any movement-related analysis.

### Neurodata without border (NWB:N) embedding

For each imaging session, imaging data, behavioral data, cell contours, cell-type prediction, calcium traces, and neuronal activity inference were combined into a single NWB:N file (*Rübel et al., 2021*). Our NWB:N data set is accessible on DANDI archive (https://gui.dandiarchive.org/#/) – ref DANDI:000219. NWB offers a common format for sharing, among others, calcium imaging data and analyzing them. Subsequently, we developed an open-source Python toolbox to analyze imaging data in the NWB format.

## Modeling

### Network implementation

We constructed a simple rate model and subsequently a more realistic spiking network in order to test our hypothesis that an increase in perisomatic inhibition could explain the switch in network dynamics between the first and second postnatal weeks. Both models consisted of one excitatory and one inhibitory population with recurrent interactions (*Figure 5A*, Appendix 1). The development of perisomatic innervation was simulated by increasing the strength from inhibitory to excitatory cells ($J_{EI}$). The external input to the model was composed of a constant and a white noise term. To estimate the responses to twitch-like inputs, an additional feedforward input composed of short pulses was fed to the network. In the rate model, the rates represented the population averaged activities. The spiking network was constructed with LIF neurons. The network connectivity was sparse and each neuron received inputs from randomly selected neurons. Presynaptic spikes resulted in exponentially decaying postsynaptic currents (see *Figure 5—source data 1* for the model parameter values). All

codes used for the modeling are available at https://gitlab.com/rouault-team-public/somatic-inhibition/ (copy archived at swh:1:rev:98b243e4bd38be6bc9addbe86bc750942cb89e21, *Hervé, 2022*) project ID: 33964849.

## Data analysis

### Sample size estimation
This study being mainly exploratory in the sense that the evolution of population activity in the CA1 region of the hippocampus during early development using large-scale imaging has not been described before, we have not been able to use explicit power calculation based on an expected size effect.

### Histological quantifications
Confocal images of synaptotagmin-2 immunostaining were analyzed using RINGO (RINGs Observation), a custom-made macro in Fiji. We first performed a max-intensity projection of the z-stack images of the top 6 µm from the slice surface, then images were cropped to restrict the analysis to the pyramidal cells layer. Obtained images were denoised using Fiji 'remove background' option and then by subtracting the mean intensity of the pixels within a manually drawn ROI in the background area (typically the cell body of a pyramidal neuron). Denoised images were then binarized using a max-entropy thresholding (Fiji option). Finally, particles with size between 0.4 µm² and 4 µm² were automatically detected using the Fiji 'Analyse particle' option. We then computed the proportion of the pyramidal cell layer (i.e., surface of the cropped region) covered by positive synaptotagmin-2 labeling.

### In vivo electrophysiology
The neuronal activity recorded from both hippocampi in vivo using a 64-channel amplifier (DIPSI, France) was analyzed post hoc. Firstly, data was downsampled to 1 kHz to save disc space. The local field potential (LFP) was band-passed (2–100 Hz) using the wavelet filter (Morlet, mother wavelet of order 6), and the common reference was subtracted to exclude the bias produced by volume conducted fluctuations of LFP. Sharp wave events (SWs) were detected using a threshold approach. Firstly, LFP was band-passed (2–45 Hz) and the difference between LFP recorded in the *strati oriens*, *pyramidale*, and *radiatum* was calculated. Events were considered as SWs if (1) LFP reversion was observed in the *stratum pyramidale* and (2) their peak amplitude in the resulting trace exceeded the threshold of 4 SDs calculated over the entire trace (the threshold corresponds to p-values<0.01). The occurrence rate of SW was calculated over the entire recording and normalized to 1 min. SW co-occurrence was also calculated by cross-correlating the SW timestamps from ipsilateral and contralateral hippocampi using a bin size of 10 ms. Spectral analysis was carried out using the Chronux toolbox. Spectral power was estimated using direct multi-taper estimators (three time-bandwidth products and five tapers).

### Statistics for in vivo electrophysiology
Group comparisons were done using nonparametric Wilcoxon rank-sum test for equal medians, and p-value of 0.05 was considered significant. Variability of the estimates was visualized as shaded bands of standard deviation computed using jackknife.

### Vigilance state determination in neonatal mice
All analyses of EMG data were completed using custom scripts in MATLAB (Cossart lab GitLab, group ID 5948056, project ID 36204799). For each experiment, the raw EMG data was first downsampled to 1000 Hz and subsequently high-pass-filtered at 300 Hz and rectified. The processed data was then plotted to allow for manual inspection. Consistent with prior reports (*Mohns and Blumberg, 2010*), the data was primarily composed of alternating periods of high EMG tone (referred to as wakefulness) associated with 'complex' movements as well as periods of low EMG tone associated with a general behavioral quiescence and the presence of periodic brief myoclonic twitches (referred to as 'active sleep' due to the frequent observation of muscle twitches; *Mohns and Blumberg, 2010*). For vigilance state determination, we therefore utilized a protocol similar to that described previously (*Del Rio-Bermudez et al., 2015*). For both the 'high' and 'low' EMG tone conditions, five periods, each 1 s in duration, were first sampled from locations spread out over the entire recording length.

Data from the samples were then pooled for each condition and the average value of the rectified signal was determined. Next, the midpoint between the average rectified signal values calculated for the 'high' and 'low' EMG tone conditions was determined for subsequent use as a threshold to separate periods of non-wakefulness (below the midpoint threshold value) from periods of wakefulness (above the midpoint threshold value), while the quarter point between these two values was calculated to further separate periods of non-wakefulness into active sleep (below the quarter point threshold value) or a sleep–wake transitory state (above the quarter point threshold value but below the midpoint threshold value). Once these thresholds were determined, the entire length of data was divided into 1 s nonoverlapping bins and the average filtered rectified EMG signal was determined for each. A hypnogram was then created by automatically applying the threshold-derived criteria to the binned averaged data. Data bins scored as being active sleep were further analyzed to determine the presence of muscle twitches; this was accomplished by automatically identifying data points with values exceeding the mean +5× standard deviation value determined from the low EMG tone representative samples. As a final step, the hypnogram and filtered rectified EMG signal data were plotted and manually inspected to ensure the accuracy of results. The hypnogram was then incorporated in the final NWB:N file to serve in the definition of the epochs of wakefulness and active sleep.

## Analysis of calcium imaging data in the NWB format using CICADA

Analysis was performed using CICADA (Cossart lab GitLab, group ID 5948056, project ID 14048984), a custom-made open-source Python toolbox allowing for the automatic analysis of calcium imaging data in the NWB:N format. CICADA offers a user-friendly graphical user interface allowing the user (1) to select the NWB files of the recording sessions to include in a given analysis, (2) select the analysis to run and set up the parameters, and (3) generate result tables and/or ready to use figures. In addition, each analysis run in CICADA generates a configuration file that can be loaded in CICADA with the option 'Load a set of parameters' allowing for the replication of the analysis. CICADA can be installed following the installation guidelines presented at https://gitlab.com/cossartlab/cicada (copy archived at swh:1:rev:2ef0c25d7da5b69849c663ed56a0033cfe8488ca; *Dard, 2022*).

## Calcium transient frequency analysis

Analysis launched from CICADA 'Transient's frequency' analysis. The transient frequency for each cell was computed using the count of calcium transient onsets divided by the duration of the recording and was then averaged across all cells imaged in one given mouse pup across one or more imaging sessions.

## SCE detection

Analysis launched from CICADA 'SCE description' analysis. SCEs were defined as the imaging frames within which the number of co-active cells exceeded the chance level as estimated using a reshuffling method. Briefly, an independent circular shift was applied to each cell to obtain 300 surrogate raster plots. We computed the 99th percentile of the distribution of the number of co-active cells from these surrogates and used this value as a threshold to define the minimal number of co-active cells in an SCE. Peak of synchrony above this threshold separated by at least five imaging frames (500 ms) was defined as SCE frames. To compute the percentage of transients within SCEs, we counted, for each cell, the number of its calcium transients (from onset to peak) crossing SCE frames and divided it by its total number of calcium transients. We averaged the obtained values over all the cells imaged per animal.

## Peri-movement time histograms (PMTH)

Analysis launched from CICADA 'Population-level PSTH' analysis. A 20-s-long time window centered on movement onset was used. For each movement within an imaging session, the number of cells activated or the sum of all cells' fluorescence was calculated for each time bin in that 20-s-long window. We obtain as many values as movements per time bin; for each individual imaging session, the 25th, median, and 75th percentiles of the distributions of these values per time bin are computed and divided by the number of imaged cells. To display the percentage of active cells at a given time bin, these values were multiplied by 100. To combine imaging sessions in an age group (i.e., P5, 6, 7, 8, 9, 10, 11, 12), all the median PMTHs from individual imaging sessions belonging to the given group

were stacked and we represented at each time bin the 25th percentile, median, and 75th percentile value of these median PMTHs. To evaluate chance level around movement onsets, 500 surrogate raster plots per imaging session were computed, and the above procedure was used to obtain chance level in each imaging session and then grouped. To obtain each surrogate raster plot, the activity of each imaged cell was translated by a randomly selected integer (between 1 and the total number of frames). We used the 95th percentile of the surrogate PMTH to conclude significant activation and the 5th percentile to conclude significant activity reduction. PMTHs obtained from fluorescence signals were built from DF/F calcium traces.

### Movement-related inhibition

Analysis launched from CICADA 'Activity ratio around epochs' analysis. A 4-s-long window centered on the onset of movements was used. The total number of cells activated during this time period was calculated. If less than 40% of these cells were activated within 2 s following movement onset, the movement was classified as an 'inhibiting' movement. This procedure was applied to all detected movements to obtain for each mouse pup the proportion of 'inhibiting' movements.

### Movement- and immobility-associated cells

Analysis launched from CICADA 'Epoch-associated cells' analysis. The number of transients per cell occurring during movement or immobility was calculated. These transient onsets were then circularly shifted 100 times and the same calculation was performed on each roll. We used the 99th percentile of this distribution as a threshold above which the cell was considered as associated with movement or immobility. Finally, the proportion of cells associated with rest or immobility was calculated for each imaged mouse.

### Statistics

Statistical tests were performed using GraphPad (Prism).

## Acknowledgements

This work was supported by the European Research Council under the European Union's, Horizon 2020 research and innovation program grant# 646925, the Fondation Bettencourt Schueller, and an NWB seed grant# R20046AA. The project leading to this publication has received funding from the «Investissements d'Avenir» French government program managed by the French National Research Agency (ANR-16-CONV-0001) and the Excellence Initiative of Aix-Marseille University – A*MIDEX. RFD was funded by the "Ministere de l'Enseignement Supérieur, de la Recherche et de l'Innovation" and the Fondation pour la Recherche Médicale Grant FDT202106012824. EL was funded by the "Ministere de l'Enseignement Supérieur, de la Recherche et de l'Innovation" and A*Midex foundation and the French National Research Agency funded by the French government «Investissements d'Avenir» program (NeuroSchool, nEURo*AMU, ANR-17-EURE-0029 grant). JD was supported by the Fondation pour la Recherche Médicale Grant FDM20170638339. MP was supported by the Fondation pour la Recherche Médicale Grant ARF20160936186. TD and TS were funded by «Investissements d'Avenir» French government program managed by the French National Research Agency (ANR-16-CONV-0001) and from the Excellence Initiative of Aix-Marseille University – A*MIDEX. We would like to thank Dr. Pierre-Pascal Lenck-Santini for providing valuable feedback on our research project. We thank S Pellegrino-Corby, F Michel, and S Brustlein from the INMED animal and imaging facilities (InMagic). We would also like to thank Marion Sicre for her help with $GAD1^{Cre/+}$ experiments. We are grateful to Pr. Hannah Monyer for providing the $GAD1^{Cre/+}$ mouse lines. We thank the Centre de Calcul Intensif d'Aix-Marseille for granting access to its high-performance computing resources. The rabies virus was a gift from Conzelman laboratory.

# Additional information

## Funding

| Funder | Grant reference number | Author |
|---|---|---|
| European Resuscitation Council | 646925 | Rosa Cossart |
| Fondation Bettencourt Schueller | | Rosa Cossart |
| Neurodata Without Borders | R20046AA | Michel A Picardo |
| Agence Nationale de la Recherche | ANR-16-CONV-0001 | Dmitrii Suchkov Rosa Cossart |
| Ministère de l'Education Nationale, de l'Enseignement Superieur et de la Recherche | MESR | Robin F Dard Erwan Leprince |
| Fondation pour la Recherche Médicale | FDT202106012824 | Robin F Dard |
| Fondation pour la Recherche Médicale | FDM20170638339 | Julien Denis |
| Fondation pour la Recherche Médicale | ARF20160936186 | Michel A Picardo |

The funders had no role in study design, data collection and interpretation, or the decision to submit the work for publication.

## Author contributions

Robin F Dard, Conceptualization, Resources, Data curation, Software, Formal analysis, Validation, Investigation, Visualization, Methodology, Writing - original draft, Writing - review and editing; Erwan Leprince, Visualization, Methodology, Writing - review and editing; Julien Denis, Conceptualization, Data curation, Software, Formal analysis, Validation, Investigation, Visualization, Methodology; Shrisha Rao Balappa, Software, Formal analysis; Dmitrii Suchkov, Formal analysis, Methodology; Richard Boyce, Data curation, Formal analysis, Methodology; Catherine Lopez, Marie Giorgi-Kurz, Methodology; Tom Szwagier, Théo Dumont, Software; Hervé Rouault, Software, Formal analysis, Validation, Visualization, Writing - review and editing; Marat Minlebaev, Formal analysis, Investigation, Visualization, Methodology, Writing - review and editing; Agnès Baude, Data curation, Investigation, Methodology, Writing - review and editing; Rosa Cossart, Conceptualization, Resources, Supervision, Funding acquisition, Validation, Investigation, Visualization, Writing - original draft, Project administration, Writing - review and editing; Michel A Picardo, Conceptualization, Resources, Data curation, Formal analysis, Supervision, Funding acquisition, Validation, Investigation, Visualization, Methodology, Writing - original draft, Project administration, Writing - review and editing

## Author ORCIDs

Julien Denis ⓘ http://orcid.org/0000-0002-0537-6483
Hervé Rouault ⓘ http://orcid.org/0000-0002-4997-2711
Marat Minlebaev ⓘ http://orcid.org/0000-0002-0722-7027
Agnès Baude ⓘ http://orcid.org/0000-0002-7025-364X
Rosa Cossart ⓘ http://orcid.org/0000-0003-2111-6638
Michel A Picardo ⓘ http://orcid.org/0000-0003-1198-3930

## Ethics

All experiments were performed under the guidelines of the French National Ethics Committee for Sciences and Health report on 'Ethical Principles for Animal Experimentation' in agreement with the European Community Directive 86/609/EEC (Apafis #18-185 and #30-959).

## Decision letter and Author response

Decision letter https://doi.org/10.7554/eLife.78116.sa1

Author response https://doi.org/10.7554/eLife.78116.sa2

## Additional files

### Supplementary files
• Transparent reporting form

• Supplementary file 1. Details of the 62 imaging sessions from the 35 mouse pups showing the number of cells recorded in each session and how they were used in the main figures. * indicates mouse pups that are used for illustration. Y, included in the panel; N, not included.

• Supplementary file 2. Details of the 62 imaging sessions from the 35 mouse pups showing the number of cells recorded in each session and how they were used in the main figures. * shows mouse pups that were used for illustration. Y, included in the panel; N, not included.

### Data availability
NWB dataset is available at DANDI Archive (https://dandiarchive.org/dandiset/000219). All codes are on GITLAB (Cossart Lab - GitLab).

The following dataset was generated:

| Author(s) | Year | Dataset title | Dataset URL | Database and Identifier |
|---|---|---|---|---|
| Robin FD | 2022 | Two photon calcium imaging in the CA1 region of the hippocampus in neonatal mice | https://gui.dandiarchive.org/#/dandiset/000219 | DANDI, 000219 |

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

# Appendix 1

Perisomatic inhibition onto pyramidal neurons in CA1 is established during the early stages of development. Firstly, we investigated the effect of perisomatic inhibition on response to twitch-like inputs in a rate model. It allowed us to analytically characterize and compare the changes in the dynamics of the model under the conditions of weak and strong inhibition. Secondly, we show through numerical simulations that similar changes can be observed in a more realistic spiking network model. We show that increasing the strength of perisomatic inhibition in our models can account for the experimentally observed decrease in responses to twitch-like feedforward inputs.

## Rate model

The rate model consists of two populations (excitatory and inhibitory) with interaction strengths $J_{ab}$, $(a, b \in E, I, J_{ab} > 0)$. They receive feedforward input of strengths $J_{a0} > 0$ from an external excitatory population with average firing rate $r_0$. The rates $r_E, r_I$ represent the population-averaged activities. The timescale of their evolution is determined by $\tau_E$ and $\tau_I$ and follows

$$\begin{pmatrix} \dot{r}_E \\ \dot{r}_I \end{pmatrix} = \begin{pmatrix} -r_E/\tau_E \\ -r_I/\tau_I \end{pmatrix} + f\left( \begin{pmatrix} J_{EE} & -J_{EI} \\ J_{IE} & -J_{II} \end{pmatrix} \begin{pmatrix} r_E \\ r_I \end{pmatrix} + \begin{pmatrix} J_{E0} \\ J_{I0} \end{pmatrix} r_0 \right) \tag{1}$$

where $f(x)$ is the neuronal transfer function. We choose it to be threshold linear, that is, $f(x) = [x]_+$. For positive $x$, the dynamics in matrix notation can be written as

$$\mathbf{\dot{r}} = (\mathbf{J} - \mathbf{I}/\tau)\mathbf{r} + \mathbf{r}_0 = \mathbf{Mr} + \mathbf{r}_0 \tag{2}$$

The fixed point of this system is given by

$$\mathbf{r}^* = -\mathbf{M}^{-1}\mathbf{r}_0 \tag{3}$$

where

$$\mathbf{M}^{-1} = \frac{1}{\det(M)} \begin{pmatrix} -J_{11} - \frac{1}{\tau_I} & J_{E1} \\ -J_{1E} & -J_{EE} + \frac{1}{\tau_E} \end{pmatrix} \tag{4}$$

$$\det(M) = J_{EI}J_{IE} - J_{EE}J_{II} + \frac{J_{11}}{\tau_E} - \frac{J_{EE}}{\tau_I} + \frac{1}{\tau_E\tau_I} \tag{5}$$

### Linear stability

The fixed point $\mathbf{r}^*$ is stable to small perturbations if the real parts of all the eigenvalues of the Jacobian matrix $\mathbf{M}(\mathbf{r}^*)$ are negative. The eigenvalues can be expressed as

$$\lambda_\pm = \frac{1}{2}\left( \mathrm{Tr}(\mathbf{M}) \pm \sqrt{\mathrm{Tr}(\mathbf{M})^2 - 4\det(M)} \right) \tag{6}$$

where

$$\mathrm{Tr}(M) = J_{EE} - J_{II} - \frac{1}{\tau_E} - \frac{1}{\tau_I} \tag{7}$$

Equivalently, the system is always stable if $\det(\mathbf{M}) > 0$ and $\mathrm{Tr}(\mathbf{M}) < 0$.
Requiring $r_e^* > 0$ and $r_i^* > 0$ gives the conditions

$$(J_{II} + \frac{1}{\tau_I})J_{E0} - J_{EI}J_{I0} > 0 \implies \frac{J_{E0}}{J_{I0}} > \frac{J_{EI}}{J_{II} + 1/\tau_I} \tag{8}$$

$$J_{IE}J_{E0} - (J_{EE} - \frac{1}{\tau_E})J_{I0} > 0 \implies \frac{J_{E0}}{J_{I0}} > \frac{J_{EE} - 1/\tau_E}{J_{IE}} \tag{9}$$

$\det(M) > 0$ gives,

$$-(J_{EE} - \frac{1}{\tau_E})(J_{II} + \frac{1}{\tau_I}) + J_{EI}J_{IE} > 0 \implies \frac{J_{EI}}{J_{II} + 1/\tau_I} > \frac{J_{EE} - 1/\tau_E}{J_{IE}} \tag{10}$$

Combining the inequalities above gives the constraints for stable nonzero rates

$$\frac{J_{E0}}{J_{I0}} > \frac{J_{EI}}{J_{II} + 1/\tau_I} > \frac{J_{EE} - 1/\tau_E}{J_{IE}} \tag{11}$$

When the solutions are stable, small perturbations $\delta\mathbf{r}$ will decay to zero. Twitches can be considered as perturbations around the fixed point. The transient response to such short impulses can be expressed as $\mathbf{r}(t) = C_1 \exp(\lambda_+) + C_2 \exp(\lambda_2)$

For a system with external white noise, $\xi(t)$ as input we have

$$\dot{\mathbf{r}} = \mathbf{M}\mathbf{r} + \mathbf{r}_0 + \sqrt{\Sigma}\,\xi(t) \tag{12}$$

$$\langle\xi(t)\rangle_t = 0, \quad \langle\xi(t)\xi(s)\rangle_t = \delta(t - s) \tag{13}$$

We set the off-diagonal elements to zero, that is, $\Sigma_{ij} = 0$ if $i \neq j$. Let $\delta\mathbf{r}(t) = \mathbf{r}(t) - \mathbf{r}^*$, so the linearized system is

$$\delta\dot{\mathbf{r}} = \mathbf{M}\,\delta\mathbf{r} + \sqrt{\Sigma}\,\xi(t)dt \tag{14}$$

This can be seen as a 2D Ornstein–Ulenbeck process defined as

$$\mathbf{x}(t) = -\mathbf{A}\,\mathbf{x}(t)\,dt + \mathbf{B}\,d\mathbf{W}(t) \tag{15}$$

which is well documented (*Gardiner and Bennett, 1985*, pp. 109–111), and we can immediately write down the expression for the covariance matrix $\mathbf{C}(\tau)$. With ($\mathbf{A} = -\mathbf{M}$), we have

$$\mathbf{C}(\tau) = \langle\delta\mathbf{r(t)}\delta\mathbf{r^T(t} + \tau)\rangle = \exp(-\mathbf{A}\,\tau)\,\sigma \tag{16}$$

$$\sigma = \frac{\det A\,\Sigma + [A - Tr(A)\mathbf{1}]\,\Sigma\,[A - Tr(A)\mathbf{1}]^T}{2\mathbf{Tr(A)}\,\mathbf{det(A)}} \tag{17}$$

## Spiking model: LIF network

Excitatory and inhibitory neurons are modeled as LIF neurons. The LIF network consists of $N_E$ excitatory neurons and $N_I$ inhibitory neurons with exponentially decaying postsynaptic currents. Each neuron receives exactly $K_E$ excitatory and $K_I$ inhibitory inputs from randomly selected neurons in the network. And we assume that the network is sparse, that is, $K \ll N$. The evolution of the subthreshold membrane voltage $V_i^a$ of neuron in population $a \in \{E, I\}$ is given by

$$\frac{dV_i^a}{dt} = -\frac{V_i^a}{\tau_m} + I_{syn} + I_{ext} + \eta, \qquad I_{syn} = \sum_{b,j} J_{ij}^{ab} S_j^{ab} \tag{18}$$

$$\frac{dS_j^{ab}}{dt} = -\frac{S_j^{ab}}{\tau_{syn}} + \sum_{t_j^b} \delta(t - t_j^b) \tag{19}$$

When the membrane voltage reaches the threshold, $V_{\text{threshold}}$, it is reset to $V_{\text{reset}}$. $\tau_{\text{syn}}$ is the synaptic time constant, and $t_j^b$ is the spike time of neuron $(i, b)$. The coupling strengths $J_{ij}^{ab} = J_{ab}$, if there is a connection from neuron $(j, b)$ to $(i, a)$ and zero otherwise. The contribution of external inputs is represented by $I_{ext}$. $I_{syn}$ represents the total synaptic currents due to spikes. Spikes are modeled as delta functions.

If a neuron $(j, b)$ emits a spike at time $t_j^b$ and projects to a postsynaptic neuron $(i, a)$, this will result in a change of the membrane voltage $V_i^a$ of the postsynaptic neuron by an amount $J_{ab}$. The membrane voltage decays exponentially to its resting potential in a time $\tau_m$. Each neuron receives an independent Gaussian white noise of amplitude $\eta$.

## Simulations and data analysis

The network simulations were conducted using custom code written in Python and C++, and all the analyses was done in Python. We use the forward Euler method to solve the set of coupled ODEs with a time step of $0.1\,\text{ms}$.

## Covariance

Given a stationary stochastic process $\mathbf{X}_t$ with mean $\mu_X = \mathbf{E}[X_t]$, the autocovarince is given by

$$C_{XX}(\tau) = \mathbf{E}\left[\left(X_t - \mu_X\right)\left(X_{t-\tau} - \mu_X\right)\right] \tag{20}$$

The covariance of $X_t$ with another process $Y_t$ is defined as

$$C_{XY}(\tau) = \mathbf{E}\left[\left(X_t - \mu_X\right)\left(Y_{t-\tau} - \mu_Y\right)\right] \tag{21}$$

