## [Editor Report]

This study investigates hippocampal dynamics over the course of early postnatal development with respect to spontaneous movements. Pioneering in vivo imaging in the hippocampus of neonatal mice, the authors find evidence for an abrupt developmental transition in this neural activity at the end of the first postnatal week in rodents and contributes to understanding how cognitive functions could emerge from the immature brain.

---

## [Decision Letter]

**Decision letter after peer review:**

Thank you for submitting your article "The rapid developmental rise of somatic inhibition disengages hippocampal dynamics from self-motion" for consideration by *eLife*. Your article has been reviewed by 3 peer reviewers, and the evaluation has been overseen by a Reviewing Editor and Laura Colgin as the Senior Editor. The following individual involved in the review of your submission has agreed to reveal their identity: Simon J B Butt (Reviewer #3).

Essential revisions:

While each reviewer has raised a number of specific concerns about the present study, there was an agreement that the following essential revisions needed to be addressed to warrant publication of the manuscript.

1. Statistical analysis needs to be improved throughout the manuscript and unified across figures. In their present form, some of the claims cannot be supported. Reviewer #1 (points #3, 4, 7) provided a detailed list of statistical tests to improve. Reviewer #2 (point #4) also raised concerns about the statistics demonstrating that PMTH differed between pyramidal and inhibitory neurons.

2. Although the method has been already published, the classification of inhibitory neurons with a deep neural network is interesting but requires more details. Specifically, there seems to be some discrepancy between ground truth and automatically labelled data (Reviewer #1, point #5; Reviewer #2, point #3) and it is unclear whether this classifier works correctly across developmental age.

3. The assessment of behavioural states must be improved especially since it is unclear how much the effects reported in the study depend or not on brain states (Reviewer #1 point #2). Furthermore, it seems that a whole class of "unclassified movements" was not used and it is unclear whether it is related to different brain states (Reviewer #3, point #3).

4. It is somehow still unclear whether the functional inhibitory inputs to pyramidal cells are changing over the ages studied as it is only really tested at P9-P10 (Reviewer #1 point #7; Reviewer #2 point #5; Reviewer #3 point #1). It would interesting to include any (even partial) data that were collected before P9. Note that this is not a request to collect more data. At any rate, some of the claims should be perhaps tempered, especially the discussion about a switch from external to internal models (Reviewer #1, point #9; Reviewer #3 point #5).

*Reviewer #1 (Recommendations for the authors):*

(1) From a methodological standpoint, sufficient data to support the conclusion that the invasive neurosurgical procedure used to perform hippocampal imaging does not disrupt network function is not presented. An estimated 7mm3 of cortical tissue of the ipsilateral somatosensory cortex is aspirated to allow insertion of the window implant. The authors do not present any data documenting the physiologic recovery of the pup from the surgery or the quantitative stability of active cells over the recording period. Such data would be important to ensure that pups of all ages had similarly recovered from the procedure and that age-dependent recovery processes do not contribute to the observed results. Furthermore, bilateral silicon probe recordings are used to suggest that ipsilateral and contralateral hippocampal network activity is comparable, but these data are difficult to interpret. Rates of eSW are presented and labeled as not statistically different between the bilateral hippocampi, but 5/6 pups show an increase in eSW rate for the contralateral hippocampus and it is concerning that the sample size is underpowered to detect what could be physiologically meaningful differences. The correlogram of eSW activity is also not centered around zero, suggesting that one hippocampus is consistently leading the other; which hippocampus (ipsilateral or contralateral) serves as the reference for this graph is not detailed in the figure or legend, but this could also indicate a difference in function of the surgically altered hippocampus. To my reading, it is also unclear what data are used for the calculation of the hippocampal power spectra, and no statistical comparison between ipsilateral and contralateral hippocampi is actually performed to support similarity. Furthermore, does aspiration of the ipsilateral somatosensory cortex affect spontaneous twitches/waking movements or the hippocampal-cortical response? For instance, is there a difference in response to body movements corresponding to the limbs contralateral to the window (cortex responding to the movements has been aspirated) vs. limbs ipsilateral to the window (cortex responding to the movements is intact)?

(2) How state-dependent dynamics are incorporated into the analysis requires further clarification. Given the state-specific activity patterns that occur in the more mature brain (e.g. hippocampal theta restricted to REM sleep and movement, sharp wave ripples in NREM and wakeful immobility) and the emergence of electrophysiologic differentiators of sleep states around P10, care should be taken to ensure that data from similar states are compared across ages. If there is a disproportionate representation of behavioral states across ages (for instance related to post-surgical comfort), sampling data across states would not control for state-dependent neural activity patterns. Specifically, the data comprising Figures 1B and 1C would be susceptible to such effects.

(3) The approach to statistical analysis of neural activity parameter changes across age is incompletely detailed and insufficient to support the authors' conclusions. The exact statistical tests, group numbers, and p-values used for data presented in Figure 1B-C, Figure 1-supplement 1A, Figure 2B-D should be detailed in the figure legends. It seems as though a linear correlation coefficient is used to quantify data in Figure 1B, but Kruskal-Wallis ANOVA between groups is used in Figure 1C. In Figure 2C, a sigmoidal fit is chosen, and in Figures 2B and 2D, no fitting is performed, though a linear relationship is mentioned for Figure 2B in the text. A unified statistical approach to modeling changes in neural properties over time should instead be used. Justification for different fits should be provided for each dataset. For example, examining the data in Figure 1C, there appears to be a decrease in the number of transients at P9-P10, which subsequently increases through P11-P12. The data appear to be underpowered for use of an 8 group ANOVA given the small number of data points per group, and potentially significant non-linearities in the number of transients over time are therefore ignored. Similarly, there appears to be a peak in the number of cells imaged at P9, but no details of which statistical testing was used are provided and again, more rigorous statistical approaches are required to understand whether significant parameter changes are occurring over time.

(4) PMTH plots require more rigorous and detailed quantitative analysis. In Figure 2A and Figure 3, these are portrayed as percentiles across the population. To properly evaluate such data, statistical methodology (for example, a shuffling procedure) should be employed to determine whether a significant modulation is present at each age and for different cell types. Modulation values could then be quantitatively compared (using statistical testing) between groups (e.g. P5-8 and P10-12 pyramidal cells and interneurons in Figure 3). Furthermore, examining the PMTH plots in Figure 2, it appears that there may be a leftward shift in the peak after P9 (i.e. increased activity occurs after movement in P5-9, but then seems more centered around the time of movement, or even before the detected movement in P10-12). If this observation were to be supported statistically, it could suggest an important change in the flow of hippocampal-cortical activity around the time of movement.

(5) The conclusions of the manuscript rely strongly on the differentiation of pyramidal cells and interneurons. Therefore, more details regarding the cell classifier (DeepCINAC) should be provided for this particular dataset. What features permit specific identification of interneurons, and are these features demonstrated to be constant across developmental age? In Figure 3-supplement 2, the average PMTHs for predicted vs. labelled interneurons are quite different; the peak percentage of active cells is ~10% for predicted interneurons, but ~25% for labeled interneurons. It also appears as though labeled interneurons have essentially no firing outside of movement-related activity in the surrounding 20-second window, in contrast to the predicted interneurons. These differences should be further explored and explained.

(6) The extra-hippocampal synaptic afferents appear to be evaluated using structural connectivity metrics, and no substantial changes were observed. However, no functional assessment of these synapses was performed. The conclusion that the mechanism of hippocampal activity changes is a result of local GABAergic innervation should be tempered given this, and more clearly explained in the discussion.

(7) In Figure 4, GABAergic axon activity was determined at P9-10 only, but this was interpreted as an increase. Without comparison to other ages, this conclusion would need to be tempered. Statistical evaluation of data in Figure 4B also does not directly support the authors' conclusions. They explain in the results text a sharp increase in labeling at P9, but their statistical testing does not show a significant difference between P7 and P9, which is the supposed inflection point. This point further illustrates the need for more rigorous and sensitive statistical testing for age-dependent changes.

(8) For the modeling data provided in Figure 5, why do the effects of inhibition persist over up to 5 seconds? Such effects would seem to extend beyond the direct effects of monosynaptic perisomatic inhibition. Most such models are presented over a span of milliseconds, not seconds. The parameters governing these timings, as well as the reasoning behind the choice of parameters used to generate the model, require justification and would improve clarity.

(9) The authors demonstrate evidence for detachment of pyramidal cell activity from movements and suggest a transition to internal representations. However, the data presented are strongly focused on peri-movement epochs, without any indication of the characteristics of neural activity during non-movement epochs. Unless the authors consider pursuing further analysis of their non-movement epoch data, implications regarding internal models and plasticity should be perhaps removed from the discussion, and this limitation acknowledged.

*Reviewer #2 (Recommendations for the authors):*

– Table 1: it is not clear why some mice were used in some figures and not in others. Some further detail would be useful here.

– Some further details on the CICADA analysis software would be very helpful, as this does not seem to have been published separately. How does this work, and why did the authors use it?

– It would be very helpful to add more details on those interneurons that were identified by the deep learning network. How do the authors justify using this method (as opposed to just looking for tdTomato labelled cells), and why was it necessary? What does 91% reliability refer to – false-positive or false-negative? The authors should provide a full breakdown of numbers of pyramidal cells misclassified as interneurons, and vice versa. Figure 3 Figure supp 2C shows a striking difference in the background activity levels of classified as opposed to labelled neurons – the authors should comment on why this is, and justify treating these cell types equally in spite of this.

– With respect to figure 3, the authors state that "the link between movement and activity evolves differentially towards the start of the second postnatal week when comparing pyramidal neurons and GABAergic interneurons, the former being inhibited or detached from movements while the latter remaining activated". Whilst this appears to be true from looking at average activity PMTH plots, there is no statistical quantification to demonstrate that this is the case. By quantifying either the peak and/or the trough of the PMTH, the authors should try to show a statistically significant interaction between how these develop with age and cell type.

– Figure 4D: in addition to the problems associated with having only one age point, this data also appears to show a striking dissociation between axonal and somatic interneuron activity: in the axonal trace (only) there is a significant dip below baseline following the movement peak. The authors should comment on the significance of this, and what it means for their conclusions.

*Reviewer #3 (Recommendations for the authors):*

1) On line 52 the authors build the argument that feedback connections are important for self-organisation of the internal state. Further, that is likely GABAergic in nature in CA1 due to the paucity of recurrent glutamatergic connections onto pyramidal projections neurons. The authors then show some evidence of the strengthening of the perisomatic connections (Figure 4), but – as they acknowledge in the discussion (lines 478-490) – could not strengthen the JEI connection as shown in their model (Figure 5A) be equally important? Is it possible to use their existing imaging data to test if more GABAergic interneurons respond to movement at P10-12 in the data shown in Figure 3? And assess the timing of the interneuron activity relative to the pyramidal cells?

(2) It would be interesting to see the primary imaging data for that shown in Figure 2A i.e. deltaF/F of cells following movement.

(3) Line 155, the authors define myoclonic twitches versus movements associated with wakefulness. In the methods (line 791), the authors further delineate "unclassified movements". Why are these not included in the results? Further, it is a shame that the authors only did one recording with nuchal EMG to better separate active sleep (AS is perhaps a better descriptor as opposed to REM) versus wakefulness. Again, it would be interesting to see the raw data from this animal.

(4) Do the authors think that the drop in peak %active cells from P8 to P9 (Figure 2A) is important? Could not a sigmoidal function be fitted to Figure 2D and if so, what would the R(2) value be?

(5) Line 368, it is not immediately clear to me why an increase in feed-forward inhibition leads to a detachment from external inputs. In primary sensory areas, an increase in feed-forward inhibition is observed in line with the emergence of fast sensory processing (see, for example, Chittajallu and Isaac, 2010).

[Editors' note: further revisions were suggested prior to acceptance, as described below.]

Thank you for resubmitting your work entitled "The rapid developmental rise of somatic inhibition disengages hippocampal dynamics from self-motion" for further consideration by *eLife*. Your revised article has been evaluated by Laura Colgin (Senior Editor) and a Reviewing Editor.

The manuscript has been improved but there are some remaining issues that need to be addressed, as outlined below:

(1) The manuscript should be clearer regarding the potential consequences of the experiments on the monitored features, especially when it comes to ipsi-contralateral differences. Reviewer #1 has raised again several concerns in this regard, from differences in electrophysiological patterns to possible differences in left-right twitches and how they specifically affect the recorded hemisphere.

(2) Reviewer #2 still has concerns regarding the statistics of PMTHs and requires further explanation.

*Reviewer #1 (Recommendations for the authors):*

The authors have addressed many of the points raised. Points requiring additional modification/clarification are documented below.

Initial Review – From a methodological standpoint, sufficient data to support the conclusion that the invasive neurosurgical procedure used to perform hippocampal imaging does not disrupt network function is not presented.

Author Response – We gently disagree with this reviewer given that, as previously indicated in the text and illustrated in Figure1 —figure supplement 1C, we had performed electrophysiological recordings in ipsi- and contralateral hippocampi and observed that hippocampal oscillations in the form of eSWs were present in both hemispheres, at several ages.

Reviewer Response – Demonstrating the presence of one type of oscillation that actually shows a trend toward a decreased occurrence rate in the surgically manipulated hemisphere, in my opinion, is not sufficient to claim that hippocampal network function is unaffected. The work that the authors perform below (with caveats also noted below), however, helps to provide support for at least a minimal disruption of the network.

Initial Review – An estimated 7mm3 of cortical tissue of the ipsilateral somatosensory cortex is aspirated to allow insertion of the window implant. The authors do not present any data documenting the physiologic recovery of the pup from the surgery or the quantitative stability of active cells over the recording period. Such data would be important to ensure that pups of all ages had similarly recovered from the procedure and that age-dependent recovery processes do not contribute to the observed results.

Author Response – We thank the reviewer for pointing out this important issue. We have performed simultaneous CA1 dynamics and EMG recordings in two animals. We found that craniotomy did not alter the structure of the sleep-wake cycle, as revealed by the quantification performed in Figure 1—figure supplement 1B. As previously described in Jouvet et al. 1967, P5-6 mice spend 70-80% of their time in active sleep, which is in agreement with our experiments. In addition, to probe the stability of active cells over our recording periods, we have computed the difference in the frequency of calcium transients between the first and last quarters of the recordings. We found that single-cell activity was stable throughout our recordings, for all ages recorded (see Author response image 1, median difference = 0,08 transient/mins).

Reviewer Response – The quantification of the sleep-wake cycle as performed in figure supplement 1B is inconsistent with currently proposed methods to do so. Indeed, the authors reference a paper from 1967 to support their work. Previously, sleep was suggested to be initiated in an undifferentiated mixed state (1,2), but more recently, quiet and active sleep is thought to be present from birth (3, 4, 5). The current figure shows "twitches", "REM", "transition" and "awake", which are not consistent with either classification scheme. This analysis should be redone in a manner consistent with a classification scheme to enable comparisons with other current literature in the field.

The quantification of the calcium transients over the course of the recording does seem convincing, and I would suggest that the authors include this figure in their supplementary data.

(1) M.G. Frank, H.C. Heller Development of REM and slow wave sleep in the rat

Am J Physiol, 272 (6 Pt 2) (1997), pp. R1792-R1799

(2) M.G. Frank, H.C. Heller The ontogeny of mammalian sleep: A reappraisal of alternative hypotheses J Sleep Res, 12 (2003), pp. 25-34

(3) A.M. Seelke, M.S. Blumberg The microstructure of active and quiet sleep as cortical δ activity emerges in infant rats Sleep, 31 (2008), pp. 691-699

(4) A.M. Seelke, K.A. Karlsson, A.J. Gall, M.S. Blumberg Extraocular muscle activity, rapid eye movements and the development of active and quiet sleep Eur J Neurosci, 22 (2005), pp. 911-920

(5) M.S. Blumberg, A.J. Gall, W.D. Todd The development of sleep-wake rhythms and the search for elemental circuits in the infant brain Behav Neurosci, 128 (2014), pp. 250-263

Initial Review – Furthermore, bilateral silicon probe recordings are used to suggest that ipsilateral and contralateral hippocampal network activity is comparable, but these data are difficult to interpret. Rates of eSW are presented and labeled as not statistically different between the bilateral hippocampi, but 5/6 pups show an increase in eSW rate for the contralateral hippocampus and it is concerning that the sample size is underpowered to detect what could be physiologically meaningful differences.

Author Response – We thank the reviewer for bringing up this important issue. Indeed, our experimental access to early hippocampal activity with 2-photon calcium imaging relies on a quite invasive procedure. However, the many control experiments we have performed indicate that early hippocampal dynamics were not significantly altered by the surgery. First, our extracellular electrophysiological recordings from a sample of 6 mice (ranging from P6 to P11, Figure 1—figure supplement 1C) show that the frequency of early sharp waves (eSW) was slightly but not significantly reduced in the ipsilateral hemisphere compared to the contralateral one. Of note, a similar “non-significant” decrease had been previously reported by another group (Graf et al. 2021 Figure S6C). As suggested by the reviewer, we can speculate that this slight decrease may result from a reduction of the sensory feedback re-afference originating from the right limbs. Indeed, we observed that movements of the right limbs (contralateral to the window implant) elicited a slightly smaller response than those from the left limbs. This observation has been added to Figure 1 – Supplement 1E and described in the results (lines 128-134) and discussion (lines 314-320).

We have performed additional control experiments using EMG nuchal electrodes in two pups aged P5 and P6. We observed that, an hour following the surgery (corresponding to the recovery time in our experimental procedure), the composition of the sleep-wake cycle (with 70 to 80 % of active sleep) was comparable to previous reports (Jouvet-Mounier, 1969, Figure 4). This quantification was added to Figure 1—figure supplement 1B (lines 82-86).

Reviewer Response – The authors have decided to acknowledge a likely difference due to reduced sensory feedback, which is reasonable, but this is not reflected in the manuscript as the statement is made "In the same way, the acute window implant did not significantly alter electrophysiological network patterns." (lines 89-90) and no mention is made in this section of the point. The Results section referenced speaks to differences in movements rather than activity patterns.

Initial Review – The correlogram of eSW activity is also not centered around zero, suggesting that one hippocampus is consistently leading the other; which hippocampus (ipsilateral or contralateral) serves as the reference for this graph is not detailed in the figure or legend, but this could also indicate a difference in the function of the surgically altered hippocampus.

Author Response – We thank the referee for making this important observation and apologize for the lack of clarity. The onsets of eSWs recorded in the ipsilateral hippocampus served as reference. As the referee noted, the two hippocampi are not perfectly synchronous with eSWs from the contralateral hippocampus leading the ipsilateral one by 12 ms. Such delay could be explained by a slight drop in local temperature due to the chamber placement as described previously for cortical up-states (Reig et al., 2010).

Reviewer Response – In the response here, the authors explain the finding by citing Reig et al., 2010, but in the manuscript, they cite this difference as being expected as per Graf et al., 2021; Valeeva et al., 2019b (lines 97-98). This is confusing and should be clarified.

Initial Review – To my reading, it is also unclear what data are used for the calculation of the hippocampal power spectra, and no statistical comparison between ipsilateral and contralateral hippocampi is actually performed to support similarity.

Author Response – We thank the reviewer for bringing up this important point. To characterize the developmental changes in the theta band power we have performed spectral analysis using the Chronux toolbox on the entire recordings (Chronux toolbox J Neurosci Methods. 2010 Sep 30;192(1):146- 51). Spectral power was estimated using direct multi-taper estimators (3 time-bandwidth products and 5 tapers). The shaded area corresponds to the confidence interval (p>0.05). As requested by this referee, we have updated the figure and added comparison between ipsi- and contralateral hippocampi. Our results demonstrated the similarity in the spectral distribution (see peak in the theta frequency band in P11 animals, which is not present in younger animals). Surprisingly, we found that the spectral power was slightly higher for the ipsilateral hippocampus.

Reviewer Response – These statistics are helpful. What is the proposed reason for this difference in spectral power? Taken together, it seems as though there are several indicators that the ipsilateral hippocampus is different in its activity than the contralateral hippocampus (trend toward different eSW rate, consistent lead/lag in regard to eSW activity, and significant differences in power spectra). These points should be mentioned in the manuscript text for transparency even if the authors argue that they do not affect the primary conclusions of the work.

Initial Review – Furthermore, does aspiration of the ipsilateral somatosensory cortex affect spontaneous twitches/waking movements or the hippocampal-cortical response? For instance, is there a difference in response to body movements corresponding to the limbs contralateral to the window (cortex responding to the movements has been aspirated) vs. limbs ipsilateral to the window (cortex responding to the movements is intact)?

Author Response – We thank the reviewer for this important comment. To address it, we have quantified in 6 imaging sessions from 5 animals (aged between P5 and P8) the CA1 response to body movements from the limbs contralateral to the imaging window (right) vs. limbs ipsilateral to the window (left). We found that movements from the right limbs elicited a slightly reduced response than the left limb movements. Regarding the number of twitches and waking movement of the right limbs compared to the left, we found fewer twitches on the contralateral side (right) as compared to the ipsilateral one (See Author response image 2). This could result from a partial lesion of the ipsilateral motor cortex in our experimental conditions. Related to this point, we would like to mention that the motor cortex, before P12, is more passively driven by spontaneous twitches than involved in driving them (Dooley Blumberg, 2018), but that up to 24 % of the twitches in P3-5 rats could already be originating from that region (An, Luhmann 2014). Our data is somehow consistent with that observation. This is now included in the manuscript but not studied any further as it is beyond the scope of the present paper.

Reviewer Response – This quantification is also informative. Although the difference is now shown in the supplementary figure, the point is to ensure that this difference does not affect the conclusions of the present study, not to study the changes themselves. For instance, it would be important to know if only the ipsilateral limb twitches are used (routed through the intact contralateral somatomotor region), is the trend in cell recruitment over development different?

Initial Review – State-dependent parameters are not adequately described, controlled, and examined quantitatively to ensure that data from similar behavioral states is being used for analysis across ages. Network activity from wakefulness, REM/active sleep, and NREM/quiet sleep should not be presumed to be indistinguishable.

Author Response – We would like to point out that our analysis across ages focused on the population response following animal movements, and not across all behavioral states. That said, it is true that two types of movements can be distinguished, namely the twitches and the complex ones. To take this behavioral heterogeneity into account, we have now separately quantified the hippocampal activation following twitches (movement during active sleep) and complex movement (during wakefulness). We show in Figure 2 —figure supplement 1B that the hippocampal response to twitches and complex movements is similar across ages. Thus, even if the amount of time spent in each behavioral state is modified over the developmental period that we have studied, we are pretty confident that it does not impact the transition we have described in the relationship between animal movements and hippocampal activity. Additionally, we were able to combine one P5 mouse pup 2p-imaging with nuchal EMG recordings and separately computed the PMTH for movements observed during REM or wakefulness (Figure 2 —figure supplement 1C). We show that CA1 hippocampal neurons were activated and time-locked to movement in both behavioral states, with only the amplitude of the population response differing between wakefulness and during REM. This point is now included in the result section (lines 148-152) and discussed (lines 324-327).

Reviewer Response – I am a bit unclear as to the conclusion here. In figure supplement 1B, the hippocampal response to twitches and complex movements is not different, but in figure supplement 1C, the hippocampal response is different between "REM" and wakefulness. Is the conclusion that the movement semiology-based analysis doesn't show differences, but the state-based analysis does (2x difference in population response), raising the concern that the semiology-based analysis proposed is not actually capturing state-based differences? The relationship between the two figure supplements should be clarified. Also, "REM" is not typically ascribed to a P5 pup (would usually be called active sleep).

Initial Review – It seems as though a linear correlation coefficient is used to quantify data in Figure 1B, but Kruskal-Wallis ANOVA between groups is used in Figure 1C.

Author Response – To address this point, we now perform a nonlinear fit instead of the Kruskal-Wallis ANOVA between groups to test for age effects in Figure 1C. Our analysis indicates a non-linear increase of the mean transient frequency over development with a local minimum at around P10. This is now included in the figure legend and results. Thanks.

Reviewer Response – This is helpful. Typically, would be appropriate to mention somewhere (perhaps in the Methods) how the fit was selected (e.g. minimizing MSE or some other method). An r2 value is not typically valid statistically for a non-linear regression (mentioned as "r2=0,30" in the figure legend at present).

Initial Review – In Figure 2C, a sigmoidal fit is chosen, and in Figures 2B and 2D, no fitting is performed, though a linear relationship is mentioned for Figure 2B in the text.

Author Response – We thank the reviewer and have now used a sigmoidal fit in Figure 2D (as in 2C) to demonstrate the increase around P9 of hippocampal activity during the period of mouse immobility. In Figure 2B we used one-way ANOVA to show a significant effect of age on the proportion of cells activated after movement and post hoc corrected multiple comparison tests to compare between age groups. Here, we show that the post-movement activity was not different for all pairs of age groups from P10 to P12, nor between P8 and P9 while all other pairs of age groups were highly significantly different. In addition, we mentioned in the text that the difference in the median between 2 consecutive age groups was similar from P5 to P9 and that the decrease was 4 times stronger between P9 and P10 than between any other pair. We have modified the text accordingly (lines 156-157).

Reviewer Response – This is helpful. Typically, would be appropriate to mention somewhere (perhaps in the Methods) how the sigmoidal fit was selected (e.g. minimizing MSE or some other method). Again, an r2 value is not typically valid statistically for non-linear regression.

Initial Review – A unified statistical approach to modeling changes in neural properties over time should instead be used. Justification for different fits should be provided for each dataset. For example, examining the data in Figure 1C, there appears to be a decrease in the number of transients at P9-P10, which subsequently increases through P11-P12. The data appear to be underpowered for use of an 8-group ANOVA given the small number of data points per group, and potentially significant non-linearities in the number of transients over time are therefore ignored. Similarly, there appears to be a peak in the number of cells imaged at P9, but no details of which statistical testing was used are provided and again, more rigorous statistical approaches are required to understand whether significant parameter changes are occurring over time.

Author Response – See response 3-b above.

Reviewer Response – See reviewer response to 3b above.

Initial Review – In Figure 4, GABAergic axon activity was determined at P9-10 only, but this was interpreted as an increase. Without comparison to other ages, this conclusion would need to be tempered.

Author Response – We thank the reviewer for this comment that reveals some lack of clarity in the previous description of our experiments. Indeed, functional GABAergic activity was also assessed before P9, however, given that there are no GABAergic axons in the CA1 pyramidal layer at early stages (for both CCK cf. Morozov and Freund 2003, and prospective PV cells cf. Figure 4A,B), there is no signal to be measured either. We have now added a new figure (Figure 4 - figure supplement 1) to clarify this point. In agreement with our Syt2 longitudinal quantification, we show, using tdTomato expression in the Gad67cre driver mouse line, that GABAergic perisomatic innervation is only visible after p9. This matches as well our attempted imaging experiments using axon enriched GCaMP in mice before P9.

As explained above, there are few, if any, GABAergic axons in the CA1 pyramidal layer before P9 (Figure 4 —figure supplement 1 specifically, supplementary video 7). There is no signal, thus comparison is pointless. We have now clarified this point (lines 242-244), thanks.

Reviewer Response – Thanks for clarifying this issue. To ensure that no pointless comparisons are inferred by the reader, it may be of benefit to change the wording of lines 255-256 from "increase in functional perisomatic GABAergic activity" to "emergence of functional perisomatic GABAergic activity."

Initial Review – Statistical evaluation of data in Figure 4B also does not directly support the authors' conclusions. They explain in the results text a sharp increase in labeling at P9, but their statistical testing does not show a significant difference between P7 and P9, which is the supposed inflection point. This point further illustrates the need for more rigorous and sensitive statistical testing for age-dependent changes.

Author Response – We had observed a significant difference between the P7 and P11 age groups (2 age groups exactly centered around P9), which demonstrates an increase in PV coverage around P9. In fact, P9 being the inflection point it is not surprising that it is neither different from P7 nor from P11 since it is precisely when axonal arborization is changing the most.

Reviewer Response – The statistical interpretation is still problematic here. If the statistical comparison between the P7 and the P9 groups is not significant, it is not accurate to report in the results that "However, around P9, a sudden increase in the density of positive labeling was observed" (lines 240-241). The reasoning about P9 being an inflection point may be the case, but the currently employed statistics do not show it. If the authors wanted to show there is an inflection point, rather than just stating what their statistics show (a significant difference between P7 and P11), a modeling approach that demonstrates a change in the curvature of the modeled fit around P9 would be appropriate.

*Reviewer #2 (Recommendations for the authors):*

Overall the authors have done a good job of responding to many of the points raised. However, I feel they have not provided a complete answer to three key points, raised by both myself and reviewer 1.

1) Statistics for PMTHs. Thanks to the authors for clarifying that there was already shuffled significance criteria in place for the PMTHs and the addition of an asterisk to the plots helps to clarify this. However, I still do not feel that the explanation as it stands is fully satisfactory.

a) It is necessary to provide more details of the shuffling procedure. The methods state simply that '500 surrogate raster plots per 770 imaging session were computed', which is insufficient detail. Unless the authors can provide a good reason otherwise, I think that the surrogates should be 20sec sections of data, drawn at random from the entire dataset, from both movement and non-movement epochs. If surrogates are not defined as above, the authors should justify why. In any case, more detail should be provided in the methods.

b) Are the 95th and 5th percentiles calculated with respect to each bin, or the entire 20s PMTH? If the former, then the authors need to account for the problem of multiple comparisons across the multiple time bins of the PMTH.

2) Statistics in addition to the PMTHs. Both Reviewer 1 and myself requested additional statistical analyses of the %cell activation data, in particular comparisons of how activation peaks and inhibition troughs evolve over development. The current argument of the authors is that it is sufficient to rely on whether PMTH activity crosses a significance threshold, or not, as a form of cross-age comparison. I don't agree with this – to take an example, the movement-activated activity peaks at P8 and P10 look visually very similar (Figure 2A). The proper statistical approach to test whether they are different is to compare them directly. Using the author's current approach, two very similar samples, not significantly different in themselves, could be judged to be from different populations based on one crossing an arbitrary shuffling-based threshold, and the other not.

The authors already have a model for better analysis in the manuscript – the analysis of activity troughs in Figure 2B. I think that a proper analysis of the data requires this method to be applied to movement-related peaks and post-movement throughs, for all data in figures 2, 3, and 4.

3) Thanks to the clarification from the authors, I now understand and accept that there are no axon terminals to image before P9. However, some more temporal precision regarding the emergence of perisomatic axon terminal activity would be helpful. The key transition dates for pyramidal cell activity run from P8 (still immature), P9 (which is transitional – no activity peak but also no trough) to P10 (mature activity). How does the emergence of axonal activity relate to this timeline? Is there a difference between P9 and P10? Is the response already mature at its first emergence (at P9)? Or does it continue to gradually increase between P9 and P12? This information would help make a more specific link between increases in perisomatic inhibition and PC activity.

Unless the authors can show these data, then phrases such as 'functional surge' should be avoided ('surge' implies a rapid maturation, which cannot the demonstrated using one time point), the authors should restrict their conclusions to stating that 'functional perisomatic activity from inhibitory interneurons can be observed at P9-10 (or similar).

---

## [Author Response]

Essential revisions:While each reviewer has raised a number of specific concerns about the present study, there was an agreement that the following essential revisions needed to be addressed to warrant publication of the manuscript.1. Statistical analysis needs to be improved throughout the manuscript and unified across figures. In their present form, some of the claims cannot be supported. Reviewer #1 (points #3, 4, 7) provided a detailed list of statistical tests to improve. Reviewer #2 (point #4) also raised concerns about the statistics demonstrating that PMTH differed between pyramidal and inhibitory neurons.

Statistical analysis is unified across figures when similar datasets (size, distribution) are being tested. We think this main point arises from the lack of clarity in our description of the statistical analysis being performed. We apologize for that and have now justified the choice for the various statistical tests employed. Statistical tests are now detailed for all figures as suggested by reviewers# 1 and 2.

2. Although the method has been already published, the classification of inhibitory neurons with a deep neural network is interesting but requires more details. Specifically, there seems to be some discrepancy between ground truth and automatically labelled data (Reviewer #1, point #5; Reviewer #2, point #3) and it is unclear whether this classifier works correctly across developmental age.

We thank the reviewers for pointing out this lack of detail. We have now clarified this point in the manuscript (see responses to Reviewer #1, point #5 and Reviewer #2, point #3).

3. The assessment of behavioural states must be improved especially since it is unclear how much the effects reported in the study depend or not on brain states (Reviewer #1 point #2). Furthermore, it seems that a whole class of "unclassified movements" was not used and it is unclear whether it is related to different brain states (Reviewer #3, point #3).

Behavioral states are now further examined in the revised manuscript (see response to Reviewer #1 point #2 and Reviewer #3, point #3). We thank the reviewers for this suggestion.

4. It is somehow still unclear whether the functional inhibitory inputs to pyramidal cells are changing over the ages studied as it is only really tested at P9-P10 (Reviewer #1 point #7; Reviewer #2 point #5; Reviewer #3 point #1). It would interesting to include any (even partial) data that were collected before P9. Note that this is not a request to collect more data. At any rate, some of the claims should be perhaps tempered, especially the discussion about a switch from external to internal models (Reviewer #1, point #9; Reviewer #3 point #5).

We directly addressed this concern. See responses to Reviewer #1 point #7, Reviewer #2 point #5; Reviewer #3 point #1 and Reviewer #1, point #9; Reviewer #3 point #5

Reviewer #1 (Recommendations for the authors):1) From a methodological standpoint, sufficient data to support the conclusion that the invasive neurosurgical procedure used to perform hippocampal imaging does not disrupt network function is not presented.

We gently disagree with this reviewer given that, as previously indicated in the text and illustrated in Figure1 —figure supplement 1C, we had performed electrophysiological recordings in ipsi and contralateral hippocampi and observed that hippocampal oscillations in the form of eSWs were present in both hemispheres, at several ages.

(1b) An estimated 7mm3 of cortical tissue of the ipsilateral somatosensory cortex is aspirated to allow insertion of the window implant. The authors do not present any data documenting the physiologic recovery of the pup from the surgery or the quantitative stability of active cells over the recording period. Such data would be important to ensure that pups of all ages had similarly recovered from the procedure and that age-dependent recovery processes do not contribute to the observed results.

We thank the reviewer for pointing out this important issue. We have performed simultaneous CA1 dynamics and EMG recordings in two animals. We found that craniotomy did not alter the structure of the sleep wake cycle, as revealed by the quantification performed in Figure 1—figure supplement 1B. As previously described in Jouvet et al. 1967, P5-6 mice spend 70-80% of their time in active sleep, which is in agreement with our experiments. In addition, to probe the stability of active cells over our recording periods, we have computed the difference in the frequency of calcium transients between the first and last quarters of the recordings. We found that single-cell activity was stable throughout our recordings, for all ages recorded (see Author response image 1, median difference = 0,08 transient/mins).

**Author response image 1. sa2fig1:** 

(1c) Furthermore, bilateral silicon probe recordings are used to suggest that ipsilateral and contralateral hippocampal network activity is comparable, but these data are difficult to interpret. Rates of eSW are presented and labeled as not statistically different between the bilateral hippocampi, but 5/6 pups show an increase in eSW rate for the contralateral hippocampus and it is concerning that the sample size is underpowered to detect what could be physiologically meaningful differences.

We thank the reviewer for bringing up this important issue. Indeed, our experimental access to early hippocampal activity with 2-photon calcium imaging relies on a quite invasive procedure. However, the many control experiments we have performed indicate that early hippocampal dynamics were not significantly altered by the surgery. First, our extracellular electrophysiological recordings from a sample of 6 mice (ranging from P6 to P11, Figure 1—figure supplement 1C) show that the frequency of early sharp waves (eSW) was slightly but not significantly reduced in the ipsilateral hemisphere compared to the contralateral one. Of note, a similar “non-significant” decrease had been previously reported by another group (Graf et al. 2021 Figure S6C). As suggested by the reviewer, we can speculate that this slight decrease may result from a reduction of the sensory feedback re-afference originating from the right limbs. Indeed, we observed that movements of the right limbs (contralateral to the window implant) elicited a slightly smaller response than those from the left limbs. This observation has been added to Figure 1 – Supplement 1E and described in the results (lines 128-134) and discussion (lines 314-320).

We have performed additional control experiments using EMG nuchal electrodes in two pups aged P5 and P6. We observed that, an hour following the surgery (corresponding to the recovery time in our experimental procedure), the composition of the sleep-wake cycle (with 70 to 80 % of active sleep) was comparable to previous reports (Jouvet-Mounier, 1969, Figure 4). This quantification was added to Figure 1—figure supplement 1B (lines 82-86).

The correlogram of eSW activity is also not centered around zero, suggesting that one hippocampus is consistently leading the other; which hippocampus (ipsilateral or contralateral) serves as the reference for this graph is not detailed in the figure or legend, but this could also indicate a difference in function of the surgically altered hippocampus.

We thank the referee for making this important observation and apologize for the lack of clarity. The onsets of eSWs recorded in the ipsilateral hippocampus served as reference. As the referee noted, the two hippocampi are not perfectly synchronous with eSWs from the contralateral hippocampus leading the ipsilateral one by 12 ms. Such delay could be explained by a slight drop in local temperature due to the chamber placement as described previously for cortical up-states (Reig et al., 2010).

To my reading, it is also unclear what data are used for the calculation of the hippocampal power spectra, and no statistical comparison between ipsilateral and contralateral hippocampi is actually performed to support similarity.

We thank the reviewer for bringing this important point. To characterize the developmental changes in the theta band power we have performed spectral analysis using the Chronux toolbox on the entire recordings (Chronux toolbox J Neurosci Methods. 2010 Sep 30;192(1):146-51). Spectral power was estimated using direct multi-taper estimators (3 time-bandwidth product and 5 tapers). Shaded area corresponds to confidence interval (p>0.05). As requested by this referee, we have updated the figure and added comparison between ipsi- and contralateral hippocampi. Our results demonstrated the similarity in the spectral distribution (see peak in theta frequency band in P11 animals, that is not present in younger animals). Surprisingly, we found that the spectral power was slightly higher for the ipsilateral hippocampus.

(1d) Furthermore, does aspiration of the ipsilateral somatosensory cortex affect spontaneous twitches/waking movements or the hippocampal-cortical response? For instance, is there a difference in response to body movements corresponding to the limbs contralateral to the window (cortex responding to the movements has been aspirated) vs. limbs ipsilateral to the window (cortex responding to the movements is intact)?

We thank the reviewer for this important comment. To address it, we have quantified in 6 imaging sessions from 5 animals (aged between P5 and P8) the CA1 response to body movements from the limbs contralateral to the imaging window (right) vs. limbs ipsilateral to the window (left). We found that movements from the right limbs elicited a slightly reduced response than the left limb movements.

Indeed, our experimental access to early hippocampal activity with 2-photon calcium imaging relies on a quite invasive procedure. However, the many control experiments we have performed indicate that early hippocampal dynamics were not significantly altered by the surgery. First, our extracellular electrophysiological recordings from a sample of 6 mice (ranging from P6 to P11, Figure 1—figure supplement 1C) show that the frequency of early sharp waves (eSW) was slightly but not significantly reduced in the ipsilateral hemisphere compared to the contralateral one. Of note, a similar “non-significant” decrease had been previously reported by another group (Graf et al. 2021 Figure S6C). As suggested by the reviewer, we can speculate that this slight decrease may result from a reduction of the sensory feedback re-afference originating from the right limbs. Indeed, we observed that movements of the right limbs (contralateral to the window implant) elicited a slightly smaller response than those from the left limbs. This observation has been added to Figure 1 – Supplement 1E and described in the results (lines 128-134) and discussion (lines 314-320).

We have performed additional control experiments using EMG nuchal electrodes in two pups aged P5 and P6. We observed that, an hour following the surgery (corresponding to the recovery time in our experimental procedure), the composition of the sleep-wake cycle (with 70 to 80 % of active sleep) was comparable to previous reports (Jouvet-Mounier, 1969, Figure 4). This quantification was added to Figure 1—figure supplement 1B (lines 82-86).

Regarding the number of twitches and waking movement of the right limbs compared to the left, we found less twitches on the contralateral side (right) as compared to the ipsilateral one (See Author response image 2 ). This could result from a partial lesion of the ipsilateral motor cortex in our experimental conditions. Related to this point, we would like to mention that the motor cortex, before P12, is more passively driven by spontaneous twitches than involved in driving them (Dooley Blumberg, 2018), but that up to 24 % of the twitches in P3-5 rats could already be originating from that region (An, Luhmann 2014). Our data is somehow consistent with that observation. This is now included in the manuscript but not studied any further as it is beyond the scope of the present paper.

(2) How state-dependent dynamics are incorporated into the analysis requires further clarification. Given the state-specific activity patterns that occur in the more mature brain (e.g. hippocampal theta restricted to REM sleep and movement, sharp wave ripples in NREM and wakeful immobility) and the emergence of electrophysiologic differentiators of sleep states around P10, care should be taken to ensure that data from similar states are compared across ages. If there is a disproportionate representation of behavioral states across ages (for instance related to post-surgical comfort), sampling data across states would not control for state-dependent neural activity patterns. Specifically, the data comprising Figures 1B and 1C would be susceptible to such effects.

We thank the reviewer for this major point. To address the heterogeneity of the movements occurring in different behavioral states, we have examined the CA1 response to twitches and complex movements occurring during REM sleep and awake states across ages, respectively and found no significant difference (Figure 2 —figure supplement 1). As a result, we decided to combine all types of movements. This is now detailed in the manuscript (lines 143-148).

(3a) The approach to statistical analysis of neural activity parameter changes across age is incompletely detailed and insufficient to support the authors' conclusions. The exact statistical tests, group numbers, and p-values used for data presented in Figure 1B-C, Figure 1-supplement 1A, Figure 2B-D should be detailed in the figure legends.

We obviously agree with this reviewer that rigorous statistics should be employed and can certify that the data analyzed in the submitted manuscript was carefully examined following that principle. We feel that his/her strong criticism regarding that point was not fully justified. In particular, we do not understand why statistical tests should be “unified” across different figures of the paper. Rather, statistical tests should be adapted to the sample size and distribution. Of course, the same tests were used for similar datasets. This revised manuscript now contains further description and justification of all the tests included in every figure panels.

We have noticed that indeed details on statistical analysis could sometimes be missing in the legends. Figure 2B-D and Figure 1-supplement 1A now mention all tests. Thanks.

(3b) It seems as though a linear correlation coefficient is used to quantify data in Figure 1B, but Kruskal-Wallis ANOVA between groups is used in Figure 1C.

To address this point, we now perform a nonlinear fit instead of the Kruskal-Wallis ANOVA between groups to test for age effects in Figure 1C. Our analysis indicates a non-linear increase of the mean transient frequency over development with a local minimum at around P10. This is now included in the figure legend and results. Thanks.

(3c) In Figure 2C, a sigmoidal fit is chosen, and in Figures 2B and 2D, no fitting is performed, though a linear relationship is mentioned for Figure 2B in the text.

We thank the reviewer and have now used a sigmoidal fit in Figure 2D (as in 2C) to demonstrate the increase around P9 of hippocampal activity during the period of mouse immobility. In Figure 2B we used one-way ANOVA to show a significant effect of age on the proportion of cells activated after movement and *post hoc* corrected multiple comparison tests to compare between age groups. Here, we show that the post movement activity was not different for all pairs of age groups from P10 to P12, nor between P8 and P9 while all other pairs of age groups were highly significantly different. In addition, we mentioned in the text that the difference in median between 2 consecutive age groups was similar from P5 to P9 and that the decrease was 4 times stronger between P9 and P10 than between any other pair. We have modified the text accordingly (lines 156-157).

(3d) A unified statistical approach to modeling changes in neural properties over time should instead be used. Justification for different fits should be provided for each dataset. For example, examining the data in Figure 1C, there appears to be a decrease in the number of transients at P9-P10, which subsequently increases through P11-P12. The data appear to be underpowered for use of an 8 group ANOVA given the small number of data points per group, and potentially significant non-linearities in the number of transients over time are therefore ignored. Similarly, there appears to be a peak in the number of cells imaged at P9, but no details of which statistical testing was used are provided and again, more rigorous statistical approaches are required to understand whether significant parameter changes are occurring over time.

See response 3b above.

(4a) PMTH plots require more rigorous and detailed quantitative analysis. In Figure 2A and Figure 3, these are portrayed as percentiles across the population. To properly evaluate such data, statistical methodology (for example, a shuffling procedure) should be employed to determine whether a significant modulation is present at each age and for different cell types.

We apologize for this confusion. We had used a shuffling procedure to assess whether post movement activation exceeded chance level. To clarify this, we have placed a star on the PMTHs illustrated in figures 2, 3, 4 when statistical significance level is reached. This procedure is presented in detail in the methods part (section Peri-Movement-Time-Histograms (PMTH)).

(4b) Modulation values could then be quantitatively compared (using statistical testing) between groups (e.g. P5-8 and P10-12 pyramidal cells and interneurons in Figure 3).

Regarding pyramidal cells at P5–8 vs P10-12: since the response at P5-8 is significantly higher than the 95th percentile from surrogate data and the response at P10-12 is significantly lower than the 5th percentile we can already conclude that the response of pyramidal cells is different between the two ages. Regarding interneurons, the point here was to say that they are significantly activated after movement in the two age groups which is shown by the fact that the response amplitude exceeds the chance level. We did not aim at comparing the modulation levels for interneurons at P5-8 compared to P10-12.

(4c) Furthermore, examining the PMTH plots in Figure 2, it appears that there may be a leftward shift in the peak after P9 (i.e. increased activity occurs after movement in P5-9, but then seems more centered around the time of movement, or even before the detected movement in P10-12). If this observation were to be supported statistically, it could suggest an important change in the flow of hippocampal-cortical activity around the time of movement.

As noticed by this reviewer, a brief increase in the proportion of active cells following movement can be seen in Figure 2 on P10-11 mouse pups. This increase does not pass the chance level and is completely suppressed in the following second leading to a significantly reduced fraction of active cells. This may be the sign of feedback inhibition. At P12, this small increase in the fraction of active cells can be observed approximately one second before the onset of movement, indicating that activity would start building up in CA1 prior to movement. A corollary discharge would increase activity prior to movement on a much shorter time scale. There is therefore no obvious explanation for this interesting phenomenon. Anticipatory cell firing prior to locomotion has been previously reported in the adult cortex (see for example Vinck et al. Neuron 2015). Different mechanisms could support such anticipatory firing, including the influence of top-down inputs, changes in arousal states or any complex neuromodulatory interactions possibly associated with changes in the sleep-wake cycle and that could involve, for example, the norepinephrine, serotonin or acetylcholine systems. Thus, this intriguing observation remains to be further explored in future studies.

(5a) The conclusions of the manuscript rely strongly on the differentiation of pyramidal cells and interneurons. Therefore, more details regarding the cell classifier (DeepCINAC) should be provided for this particular dataset. What features permit specific identification of interneurons, and are these features demonstrated to be constant across developmental age?

We thank the reviewer for bringing up this important point. All the details regarding our cell type classifier can be found in the publication cited (Denis et al. 2020) as well as online on our gitlab account (https://gitlab.com/cossartlab/deepcinac). The specific dataset used is this paper is actually the one we used to train and validate the DeepCINAC cell type classifier across ages from P5 to P12; thus, the performance of the classifier in terms of sensitivity, precision and F1 score is exactly as described in (Denis et al). The features used by the classifier to perform are not easily accessible from the artificial network and we have not investigated this specific question since the performance at the classification was 91 % precision for interneurons. The input to the neuronal network is a 100 frames long movie of 25 × 25 pixels window centered on the cell body of interest. Thus, we hypothesize that the network would use directly from movie visualization several parameters such as: neuronal shape, localization and activity. Whether all, several or only one parameter has been used by the network remains an open question.

(5b) In Figure 3-supplement 2, the average PMTHs for predicted vs. labelled interneurons are quite different; the peak percentage of active cells is ~10% for predicted interneurons, but ~25% for labeled interneurons. It also appears as though labeled interneurons have essentially no firing outside of movement-related activity in the surrounding 20-second window, in contrast to the predicted interneurons. These differences should be further explored and explained.

The difference in the peak percentage of active cells is simply a direct consequence of the relatively low number of labeled interneurons in each imaging session. For example, in an imaging session with a low number of labeled interneurons the median percentage of active labeled interneurons can only take discrete values, sometimes up to 100 %. Because we then represented for each time bin the median from each session median it could bias the percentage of active labeled interneurons toward low baseline and high response. To circumvent this limitation, we represent the activation of labeled interneurons using the median DF/F signal that is more representative in case of a low number of cells. The point of Figure 3 —figure supplement 1E was to show that labeled interneurons and inferred interneurons have the same relationship with movement. This is confirmed by the PMTH using the DF/F signal from these 2 groups.

(6) The extra-hippocampal synaptic afferents appear to be evaluated using structural connectivity metrics, and no substantial changes were observed. However, no functional assessment of these synapses was performed. The conclusion that the mechanism of hippocampal activity changes is a result of local GABAergic innervation should be tempered given this, and more clearly explained in the discussion.

We agree with reviewer 1 that we did not functionally assess the extrinsic inputs, given the experimental challenge to perform such experiments. This was acknowledged in the first section of the discussion (line 301) “This [activity change] is likely due to the time-locked anatomical and functional rise of somatic GABAergic activity given that interneurons remain highly active throughout this period, including in response to spontaneous movements.” We therefore had already tempered the conclusion that the mechanism of hippocampal activity changes is a result of local GABAergic innervation. This conclusion is softened even further in the discussion of the revised manuscript. That said, our modeling approach further supports the possibility that a rise in local GABAergic inhibition alone can account for the changes in CA1 dynamics reported here.

(7a) In Figure 4, GABAergic axon activity was determined at P9-10 only, but this was interpreted as an increase. Without comparison to other ages, this conclusion would need to be tempered.

We thank the reviewer for this comment that reveals some lack of clarity in the previous description of our experiments. Indeed, functional GABAergic activity was also assessed before P9, however, given that there are no GABAergic axons in the CA1 pyramidal layer at early stages (for both CCK cf. Morozov and Freund 2003, and prospective PV cells cf. Figure 4A,B), there is no signal to be measured either. We have now added a new figure (Figure 4 - figure supplement 1) to clarify this point. In agreement with our Syt2 longitudinal quantification, we show, using tdTomato expression in the Gad67cre driver mouse line, that GABAergic perisomatic innervation is only visible after p9. This matches as well, our attempted imaging experiments using axon enriched GCaMP in mice before P9.

As explained above, there are few, if any, GABAergic axons in the CA1 pyramidal layer before P9 (Figure 4 —figure supplement 1 specifically, Video 7). There is no signal, thus comparison is pointless. We have now clarified this point (lines 242-244), thanks.

(7b) Statistical evaluation of data in Figure 4B also does not directly support the authors' conclusions. They explain in the results text a sharp increase in labeling at P9, but their statistical testing does not show a significant difference between P7 and P9, which is the supposed inflection point. This point further illustrates the need for more rigorous and sensitive statistical testing for age-dependent changes.

We had observed a significant difference between the P7 and P11 age groups (2 age groups exactly centered around P9), which demonstrates an increase of PV coverage around P9. In fact, P9 being the inflexion point it is not surprising that it is neither different from P7 nor from P11 since it is precisely when axonal arborization is changing the most.

(8) For the modeling data provided in Figure 5, why do the effects of inhibition persist over up to 5 seconds? Such effects would seem to extend beyond the direct effects of monosynaptic perisomatic inhibition. Most such models are presented over a span of milliseconds, not seconds. The parameters governing these timings, as well as the reasoning behind the choice of parameters used to generate the model, require justification and would improve clarity.

We agree that the time scales used in our simulations need clarification. The fast excitatory and inhibitory synaptic time scales fall beyond the time resolution offered by GCamp6s. It is believed that they tend to produce irregular spiking as demonstrated by the balanced state models (van Vreeswijk and Sompolinsky, 1996). In our simulation of the leaky integrate and fire network, we account for this irregular spiking by providing a noisy, normally distributed, input to all the cells.

We believe the longer observed time scales (in the range of a few hundreds of ms to seconds) originate from slow synaptic transmission. In order to make that more precise, we have updated the synaptic time scales employed in our simulations to match those receptors (Destexhe, Mainen, and Sejnowski, 1994) and generated new panels accordingly in figure 5. This is now better clarified in the manuscript (lines 260-264).

(9) The authors demonstrate evidence for detachment of pyramidal cell activity from movements and suggest a transition to internal representations. However, the data presented are strongly focused on peri-movement epochs, without any indication of the characteristics of neural activity during non-movement epochs. Unless the authors consider pursuing further analysis of their non-movement epoch data, implications regarding internal models and plasticity should be perhaps removed from the discussion, and this limitation acknowledged.

We thank the reviewer for his/her helpful comment. In the abstract and discussion parts we have removed the statements regarding ‘internal models’.

Reviewer #2 (Recommendations for the authors):– Table 1: it is not clear why some mice were used in some figures and not in others. Some further detail would be useful here.

We apologize for the lack of details regarding the criteria for inclusion of mice from imaging experiments. Due to technical problems during the recording of the mouse behavior (behavioral frames dropped at unknown timestamps) we were not able in some cases to realign mouse behavior with calcium imaging. We decided to exclude these imaging sessions from all movement-related analysis. This explains why, for instance, some animals from figure 1 were not included in figure 2. To clarify this point, we have added a sentence in the Methods section (DATA PREPROCESSING, behavior lines 535-538).

– Some further details on the CICADA analysis software would be very helpful, as this does not seem to have been published separately. How does this work, and why did the authors use it?

We agree with reviewer #2 and added further details on the CICADA analysis pipeline in the methods section (Data Analysis, Analysis of calcium imaging data in the NWB format using CICADA lines 722-730)

– It would be very helpful to add more details on those interneurons that were identified by the deep learning network. How do the authors justify using this method (as opposed to just looking for tdTomato labelled cells), and why was it necessary?

This method allows us to infer neuronal cell types (Interneurons or Pyramidal Cells) in wild type animals. This is useful and necessary as not all imaged mice were from transgenic animals with tdTomato expressed in GABAergic neurons.

What does 91% reliability refer to – false-positive or false-negative? The authors should provide a full breakdown of numbers of pyramidal cells misclassified as interneurons, and vice versa. Figure 3 Figure supp 2C shows a striking difference in the background activity levels of classified as opposed to labelled neurons – the authors should comment on why this is, and justify treating these cell types equally in spite of this.

We apologize for the confusion. 91% refers to the specificity of the classifier meaning the probability of being a GABAergic neuron when the classifier predicts it to be the case. This reviewer is concerned by the fact that the PMTHs plotting the proportion of active interneurons differ when they are labeled vs. inferred. The difference in background activity levels likely originates from the low number of labeled interneurons per imaging session. The point of Figure 3 —figure supplement 1E was to show that labeled and inferred interneurons had the same relationship with movement (i.e. an increased activation after movement). This is now further supported by a PMTH using the raw DF/F signals from these 2 groups (which are less sensitive to the number of cells). More details on the cell type classifier and on the identified interneurons are provided in Figure 3 —figure supplement 1.

– With respect to figure 3, the authors state that "the link between movement and activity evolves differentially towards the start of the second postnatal week when comparing pyramidal neurons and GABAergic interneurons, the former being inhibited or detached from movements while the latter remaining activated". Whilst this appears to be true from looking at average activity PMTH plots, there is no statistical quantification to demonstrate that this is the case. By quantifying either the peak and/or the trough of the PMTH, the authors should try to show a statistically significant interaction between how these develop with age and cell type.

There was actually a statistical quantification included in the PMTH plots. We used surrogates of neuronal activity to estimate the chance level. We showed that PMTH peaks in P5-8 mice are above the chance level for both interneurons and pyramidal cells, indicating significant co-activation, whereas in P10-12 mice PMTH peaks are below the chance level for pyramidal cells and remain above chance for interneurons. We did not specifically quantify differences in peak or trough values but rather demonstrated that these peaks/troughs are significantly above/below chance levels. This shows that “the link between movement and activity evolves differentially towards the start of the second postnatal week when comparing pyramidal neurons and GABAergic interneurons, the former being inhibited or detached from movements while the latter remaining activated". To further clarify this point, we now indicate with a star on the PMTHs from figures 2, 3, 4 and 5, cases when statistical significance level is exceeded. This analysis procedure is presented in detail in the methods part (section Peri-Movement-Time-Histograms (PMTH)).

– Figure 4D: in addition to the problems associated with having only one age point, this data also appears to show a striking dissociation between axonal and somatic interneuron activity: in the axonal trace (only) there is a significant dip below baseline following the movement peak. The authors should comment on the significance of this, and what it means for their conclusions.

As explained above, we have imaged activity in GABAergic axons only after P9, not due to a technical problem but rather because there is no signal to measure given that perisomatic innervation is absent before P9. We have dedicated a new supplementary figure (Figure 4 —figure supplement 1) to explain why we could not image GABAergic axons in the pyramidal cell layer at earlier developmental stages.

The apparent dissociation between axonal and somatic interneuron activity is in fact due to the method used to build the PMTHs. PMTHs on somatic interneuron activation (Figure 2-3) are built on inferred activity from DeepCICNAC classifiers and represent the fraction of active cells across time. The PMTH in Figure 4 corresponds to the median DF/F signal centered on the onset of movement. It is not expected that these two PMTH should have the exact same shape.

We now also provide somatic PMTHs using DF/F signals. Figure 3 —figure supplement 2B illustrates the somatic response of interneurons to movement at P10-12. This time, the shape of the PMTH nicely matches the one observed for axonal imaging in Figure 4D. We thank the reviewer for pointing out this possible source of confusion.

Reviewer #3 (Recommendations for the authors):(1) On line 52 the authors build the argument that feedback connections are important for self-organisation of the internal state. Further, that is likely GABAergic in nature in CA1 due to the paucity of recurrent glutamatergic connections onto pyramidal projections neurons. The authors then show some evidence of the strengthening of the perisomatic connections (Figure 4), but – as they acknowledge in the discussion (lines 478-490) – could not strengthen the JEI connection as shown in their model (Figure 5A) be equally important? Is it possible to use their existing imaging data to test if more GABAergic interneurons respond to movement at P10-12 in the data shown in Figure 3?

We thank the reviewer for an excellent suggestion. Using our dataset, we have quantified the differences between the peak and baseline values in the PMTHs of active interneurons at P5-8 and P10-12 and found no significant difference (P5-8=5.7%; P10-12=6.1%). This suggests that the same amount of interneurons relative to baseline were recruited when the animal was moving at both stages.

And assess the timing of the interneuron activity relative to the pyramidal cells?

The timing of interneurons activity relative to pyramidal cells is visible in the cross-correlogram plots (Figure 5C). It shows that at P9-12, pyramidal cells are inhibited following interneurons activation.

(2) It would be interesting to see the primary imaging data for that shown in Figure 2A i.e. deltaF/F of cells following movement.

We agree with the reviewer and now provide the PMTHs built using the DF/F signal in addition to the ones built using inferred activity, whenever possible (Figures 2,3). These results are now illustrated in Figure 2 Figure supplement 1, Figure 3 Figure supplement 1 and Figure 3 Figure supplement 2.

(3) Line 155, the authors define myoclonic twitches versus movements associated with wakefulness. In the methods (line 791), the authors further delineate "unclassified movements". Why are these not included in the results?

We apologize for this lack of clarity. Movements referred to as ‘unclassified’ were indeed included in the main analysis. They were only excluded when analysis was restricted to ‘twitches’ or to ‘complex movements’. We have now clarified the definition of movements in the methods section (DATA PREPROCESSING, behavior lines 636-643)

Further, it is a shame that the authors only did one recording with nuchal EMG to better separate active sleep (AS is perhaps a better descriptor as opposed to REM) versus wakefulness. Again, it would be interesting to see the raw data from this animal.

Combining nuchal EMG and 2p imaging in young mice is extremely difficult. As a result, we had to perform 8 experiments for one animal meeting our standard criteria for EMG and imaging experiments. To address reviewer #3 concern, we now present the result obtained on the raw DF/F for this mouse (Figure 2 Figure supplement 1C).

(4) Do the authors think that the drop in peak %active cells from P8 to P9 (Figure 2A) is important?

Yes, we think that this drop in the peak of percentage of active cells is important. CA1 dynamics switch from being mainly driven by bottom up inputs to be inhibited by these inputs. Interestingly, in adults it is known that again CA1 activity increases when the mouse moves. There is therefore a transient inhibitory relationship between movement and hippocampal activity that is likely to carry a developmental function.

(4b) Could not a sigmoidal function be fitted to Figure 2D and if so, what would the R(2) value be?

We would like to thank this reviewer for an excellent suggestion. We were indeed able to fit a sigmoidal function to the data presented in Figure 2D (r2 = 0.55).

(5) Line 368, it is not immediately clear to me why an increase in feed-forward inhibition leads to a detachment from external inputs. In primary sensory areas, an increase in feed-forward inhibition is observed in line with the emergence of fast sensory processing (see, for example, Chittajallu and Isaac, 2010).

We agree with this reviewer that “detachment from external inputs” might not be the only interpretation of our observations. We used this terminology, inspired by this review (Buzsaki et al. Emergence of cognition from action, 2014) and also because, as discussed above, we know that P9 opens a transient period of quiescence in CA1 with locomotion, in contrast to sensory processing in cortical areas, which remains under the tight control of inhibition.

[Editors' note: further revisions were suggested prior to acceptance, as described below.]

The manuscript has been improved but there are some remaining issues that need to be addressed, as outlined below:(1) The manuscript should be clearer regarding the potential consequences of the experiments on the monitored features, especially when it comes to ipsi-contralateral differences. Reviewer #1 has raised again several concerns in this regard, from differences in electrophysiological patterns to possible differences in left-right twitches and how they specifically affect the recorded hemisphere.(2) Reviewer #2 still has concerns regarding the statistics of PMTHs and requires further explanation.Reviewer #1 (Recommendations for the authors):The authors have addressed many of the points raised. Points requiring additional modification/clarification are documented below.1. Initial Review – From a methodological standpoint, sufficient data to support the conclusion that the invasive neurosurgical procedure used to perform hippocampal imaging does not disrupt network function is not presented.Author Response – We gently disagree with this reviewer given that, as previously indicated in the text and illustrated in Figure1 —figure supplement 1C, we had performed electrophysiological recordings in ipsi- and contralateral hippocampi and observed that hippocampal oscillations in the form of eSWs were present in both hemispheres, at several ages.Reviewer Response – Demonstrating the presence of one type of oscillation that actually shows a trend toward a decreased occurrence rate in the surgically manipulated hemisphere, in my opinion, is not sufficient to claim that hippocampal network function is unaffected. The work that the authors perform below (with caveats also noted below), however, helps to provide support for at least a minimal disruption of the network.

We agree with reviewer #1 and modified the manuscript accordingly. We thus reworded line 104 “the window implant preserved the electrophysiological network patterns” by “the window implant minimally disrupted the electrophysiological network patterns”. We also added in the Results section line 92-94: “This slight and non-significant reduction in eSW frequency in the ipsilateral hemisphere was comparable with previous study using the same surgical approach (see Discussion, Graf et al., 2021).”

2. Initial Review – An estimated 7mm3 of cortical tissue of the ipsilateral somatosensory cortex is aspirated to allow insertion of the window implant. The authors do not present any data documenting the physiologic recovery of the pup from the surgery or the quantitative stability of active cells over the recording period. Such data would be important to ensure that pups of all ages had similarly recovered from the procedure and that age-dependent recovery processes do not contribute to the observed results.Author Response – We thank the reviewer for pointing out this important issue. We have performed simultaneous CA1 dynamics and EMG recordings in two animals. We found that craniotomy did not alter the structure of the sleep-wake cycle, as revealed by the quantification performed in Figure 1—figure supplement 1B. As previously described in Jouvet et al. 1967, P5-6 mice spend 70-80% of their time in active sleep, which is in agreement with our experiments. In addition, to probe the stability of active cells over our recording periods, we have computed the difference in the frequency of calcium transients between the first and last quarters of the recordings. We found that single-cell activity was stable throughout our recordings, for all ages recorded (see Author response image 1, median difference = 0,08 transient/mins).Reviewer Response – The quantification of the sleep-wake cycle as performed in figure supplement 1B is inconsistent with currently proposed methods to do so. Indeed, the authors reference a paper from 1967 to support their work. Previously, sleep was suggested to be initiated in an undifferentiated mixed state (1,2), but more recently, quiet and active sleep is thought to be present from birth (3, 4, 5). The current figure shows "twitches", "REM", "transition" and "awake", which are not consistent with either classification scheme. This analysis should be redone in a manner consistent with a classification scheme to enable comparisons with other current literature in the field.

We thank the reviewer for his/her comment. As presented in the method section (lines 708-732) the classification of animal states was similar to the one used in Rio-

Bermudez et al. 2015. To perfectly match this nomenclature, we renamed the “transition” state to “quiet sleep” (Figure 1 —figure supplement 1B). Of note, twitches were not defined as a stage but just detected events during REM/active sleep (lines 729-730).

The quantification of the calcium transients over the course of the recording does seem convincing, and I would suggest that the authors include this figure in their supplementary data.

We have now included this figure in Figure 1 —figure supplement 1E and mention it in the Results section lines 115-116 : “Neuronal activity was stable over the duration of the recording (Figure 1 —figure supplement 1D, E – median change 0.08 transients/minute, N=31 pups).”

3. Initial Review – Furthermore, bilateral silicon probe recordings are used to suggest that ipsilateral and contralateral hippocampal network activity is comparable, but these data are difficult to interpret. Rates of eSW are presented and labeled as not statistically different between the bilateral hippocampi, but 5/6 pups show an increase in eSW rate for the contralateral hippocampus and it is concerning that the sample size is underpowered to detect what could be physiologically meaningful differences.Author Response – This question on the rate of eSW raised here by the reviewer was addressed in the public review #1.Reviewer Response – The authors have decided to acknowledge a likely difference due to reduced sensory feedback, which is reasonable, but this is not reflected in the manuscript as the statement is made "In the same way, the acute window implant did not significantly alter electrophysiological network patterns." (lines 89-90) and no mention is made in this section of the point. The Results section referenced speaks to differences in movements rather than activity patterns.

We thank reviewer #1 for his/her comment (see first Author re-Response). We now clearly acknowledge these differences in the discussion part (lines 323-330): “Of note, it is important to keep in mind that part of the overlying cortex, including the primary sensory cortex, was removed to grant optical access to the hippocampus. This region may contribute in relaying the sensory feedback from the twitches to the hippocampus (Khazipov and Milh, 2018; Valeeva et al., 2019a). The surgical procedure may thus damage incoming axons from the temporoammonic track linking the entorhinal cortex to the hippocampus. Accordingly, we observed: (i) that the CA1 response to movements from the contralateral limbs was slightly reduced; (ii) that the eSW frequency in the ipsilateral hemisphere was slightly diminished; (iii) a small increase in the power spectra of network oscillations below 20Hz”

4. Initial Review – The correlogram of eSW activity is also not centered around zero, suggesting that one hippocampus is consistently leading the other; which hippocampus (ipsilateral or contralateral) serves as the reference for this graph is not detailed in the figure or legend, but this could also indicate a difference in the function of the surgically altered hippocampus.Author Response – We thank the referee for making this important observation and apologize for the lack of clarity. The onsets of eSWs recorded in the ipsilateral hippocampus served as reference. As the referee noted, the two hippocampi are not perfectly synchronous with eSWs from the contralateral hippocampus leading the ipsilateral one by 12 ms. Such delay could be explained by a slight drop in local temperature due to the chamber placement as described previously for cortical up-states (Reig et al., 2010).Reviewer Response – In the response here, the authors explain the finding by citing Reig et al., 2010, but in the manuscript, they cite this difference as being expected as per Graf et al., 2021; Valeeva et al., 2019b (lines 97-98). This is confusing and should be clarified.

We clarified this point (lines 92-97), thank you. (i) Reference Graf et al., refers to the expected slight decrease in the eSW frequency in the ipsilateral hemisphere. (ii) Reference Valeeva et al., refers to the expected synchrony in the eSW in the two hemispheres. (iii) Reference Reig et al., may provide an explanation for the delay that we observed in our experiments.

5. Initial Review – To my reading, it is also unclear what data are used for the calculation of the hippocampal power spectra, and no statistical comparison between ipsilateral and contralateral hippocampi is actually performed to support similarity.Author Response – We thank the reviewer for bringing up this important point. To characterize the developmental changes in the theta band power we have performed spectral analysis using the Chronux toolbox on the entire recordings (Chronux toolbox J Neurosci Methods. 2010 Sep 30;192(1):146- 51). Spectral power was estimated using direct multi-taper estimators (3 time-bandwidth products and 5 tapers). The shaded area corresponds to the confidence interval (p>0.05). As requested by this referee, we have updated the figure and added comparison between ipsi- and contralateral hippocampi. Our results demonstrated the similarity in the spectral distribution (see peak in the theta frequency band in P11 animals, which is not present in younger animals). Surprisingly, we found that the spectral power was slightly higher for the ipsilateral hippocampus.Reviewer Response – These statistics are helpful. What is the proposed reason for this difference in spectral power? Taken together, it seems as though there are several indicators that the ipsilateral hippocampus is different in its activity than the contralateral hippocampus (trend toward different eSW rate, consistent lead/lag in regard to eSW activity, and significant differences in power spectra). These points should be mentioned in the manuscript text for transparency even if the authors argue that they do not affect the primary conclusions of the work.

We agree with reviewer #1 and have now mentioned all these points in the manuscript (see discussion, lines 323-330, results line 102).

6. Initial Review – Furthermore, does aspiration of the ipsilateral somatosensory cortex affect spontaneous twitches/waking movements or the hippocampal-cortical response? For instance, is there a difference in response to body movements corresponding to the limbs contralateral to the window (cortex responding to the movements has been aspirated) vs. limbs ipsilateral to the window (cortex responding to the movements is intact)?Author Response – We thank the reviewer for this important comment. To address it, we have quantified in 6 imaging sessions from 5 animals (aged between P5 and P8) the CA1 response to body movements from the limbs contralateral to the imaging window (right) vs. limbs ipsilateral to the window (left). We found that movements from the right limbs elicited a slightly reduced response than the left limb movements. Regarding the number of twitches and waking movement of the right limbs compared to the left, we found fewer twitches on the contralateral side (right) as compared to the ipsilateral one (See Author response image 2). This could result from a partial lesion of the ipsilateral motor cortex in our experimental conditions. Related to this point, we would like to mention that the motor cortex, before P12, is more passively driven by spontaneous twitches than involved in driving them (Dooley Blumberg, 2018), but that up to 24 % of the twitches in P3-5 rats could already be originating from that region (An, Luhmann 2014). Our data is somehow consistent with that observation. This is now included in the manuscript but not studied any further as it is beyond the scope of the present paper.Reviewer Response – This quantification is also informative. Although the difference is now shown in the supplementary figure, the point is to ensure that this difference does not affect the conclusions of the present study, not to study the changes themselves. For instance, it would be important to know if only the ipsilateral limb twitches are used (routed through the intact contralateral somatomotor region), is the trend in cell recruitment over development different?

We have performed an additional set of analyses that confirms that the trend in the cell recruitment over development is similar when considering separately ipsi (left) and contra-lateral (right) body twitches, with the emergence of inhibition around P9 (Author response image 3) . For this, we used the already annotated twitches for two P5 and P7 pups and manually annotated randomly selected twitches in one P9 and one P12 mouse. Of note the manual annotation of all of the twitches at P9 and 12 would take several weeks. We hope that this reviewer will now be convinced that the trend reported in Figure 2A with combined twitches is also valid when these are treated separately.

**Author response image 3. sa2fig3:** PMTHs for P5, 7, 9 and P12 pups built on twitches only. N represents the number of animals. n represents the number of imaging sessions.

7. Initial Review – State-dependent parameters are not adequately described, controlled, and examined quantitatively to ensure that data from similar behavioral states is being used for analysis across ages. Network activity from wakefulness, REM/active sleep, and NREM/quiet sleep should not be presumed to be indistinguishable.Author Response – We would like to point out that our analysis across ages focused on the population response following animal movements, and not across all behavioral states. That said, it is true that two types of movements can be distinguished, namely the twitches and the complex ones. To take this behavioral heterogeneity into account, we have now separately quantified the hippocampal activation following twitches (movement during active sleep) and complex movement (during wakefulness). We show in Figure 2 —figure supplement 1B that the hippocampal response to twitches and complex movements is similar across ages. Thus, even if the amount of time spent in each behavioral state is modified over the developmental period that we have studied, we are pretty confident that it does not impact the transition we have described in the relationship between animal movements and hippocampal activity. Additionally, we were able to combine one P5 mouse pup 2p-imaging with nuchal EMG recordings and separately computed the PMTH for movements observed during REM or wakefulness (Figure 2 —figure supplement 1C). We show that CA1 hippocampal neurons were activated and time-locked to movement in both behavioral states, with only the amplitude of the population response differing between wakefulness and during REM. This point is now included in the result section (lines 148-152) and discussed (lines 324-327).Reviewer Response – I am a bit unclear as to the conclusion here. In figure supplement 1B, the hippocampal response to twitches and complex movements is not different, but in figure supplement 1C, the hippocampal response is different between "REM" and wakefulness. Is the conclusion that the movement semiology-based analysis doesn't show differences, but the state-based analysis does (2x difference in population response), raising the concern that the semiology-based analysis proposed is not actually capturing state-based differences? The relationship between the two figure supplements should be clarified. Also, "REM" is not typically ascribed to a P5 pup (would usually be called active sleep).

We apologize for the confusion in this figure. It is true that Figure 2 —figure supplement 1B and 1C display some differences in the cell recruitment during twitching and in active/REM sleep movements that are mainly twitches as well. This can be explained by the number of animals used to build each figure. For figure 2 —figure supplement 1B we had pooled multiple animals (e.g., P5 represents n=4 pups and 8 imaging sessions). However, Figure 2 —figure supplement 1C shows the PMTHs for REM and Awake only for one P5 mouse pup.

In Author response image 4 we confirm that comparing twitches and complex movements gives the same result as comparing REM/active sleep movements and awake movements in the same mouse pup compare Figure 2 —figure supplement 1C top panels with (Author response image 4). This confirms that the twitches and movements identified by manual annotation of the video recording captures a state-based difference between REM and AWAKE movement when there is one.

**Author response image 4. sa2fig4:** PMTHs for P5 pup on twitches and complex movements (same pup as the one used in Figure 2 —figure supplement 1C to compare REM/ active sleep movements and awake movements).

Additionally, we have now clarified the manuscript (see lines 152-158): “Accordingly, when combining calcium imaging with nuchal EMG recordings in one P5 mouse pup, we observed an increase in the percentage of active cells and in the DF / F fluorescence signal following movements occurring both during REM sleep and wakefulness (Figure 2 —figure supplement 1C). However, when combining all mouse pups, and considering separately twitches (occurring during REM/active sleep) and complex movements (occurring during wakefulness), based on video recordings, we found that the two movement types did not significantly differ in their impact on CA1 activity (Figure 2 —figure supplement 1B)”.

8. Initial Review – It seems as though a linear correlation coefficient is used to quantify data in Figure 1B, but Kruskal-Wallis ANOVA between groups is used in Figure 1C.Author Response – To address this point, we now perform a nonlinear fit instead of the Kruskal-Wallis ANOVA between groups to test for age effects in Figure 1C. Our analysis indicates a non-linear increase of the mean transient frequency over development with a local minimum at around P10. This is now included in the figure legend and results. Thanks.Reviewer Response – This is helpful. Typically, would be appropriate to mention somewhere (perhaps in the Methods) how the fit was selected (e.g. minimizing MSE or some other method). An r2 value is not typically valid statistically for a non-linear regression (mentioned as "r2=0,30" in the figure legend at present).

The fourth order polynomial fit in Figure 1C uses least squares method. This is now mentioned in the figure legend (line 808).

9. Initial Review – In Figure 2C, a sigmoidal fit is chosen, and in Figures 2B and 2D, no fitting is performed, though a linear relationship is mentioned for Figure 2B in the text.Author Response – We thank the reviewer and have now used a sigmoidal fit in Figure 2D (as in 2C) to demonstrate the increase around P9 of hippocampal activity during the period of mouse immobility. In Figure 2B we used one-way ANOVA to show a significant effect of age on the proportion of cells activated after movement and post hoc corrected multiple comparison tests to compare between age groups. Here, we show that the post-movement activity was not different for all pairs of age groups from P10 to P12, nor between P8 and P9 while all other pairs of age groups were highly significantly different. In addition, we mentioned in the text that the difference in the median between 2 consecutive age groups was similar from P5 to P9 and that the decrease was 4 times stronger between P9 and P10 than between any other pair. We have modified the text accordingly (lines 156-157).Reviewer Response – This is helpful. Typically, would be appropriate to mention somewhere (perhaps in the Methods) how the sigmoidal fit was selected (e.g. minimizing MSE or some other method). Again, an r2 value is not typically valid statistically for non-linear regression.

The sigmoïdal fit in Figure 2C,D uses least squares method. This is now mentioned in the figure legend (lines 837 and 842).

10. Initial Review – A unified statistical approach to modeling changes in neural properties over time should instead be used. Justification for different fits should be provided for each dataset. For example, examining the data in Figure 1C, there appears to be a decrease in the number of transients at P9-P10, which subsequently increases through P11-P12. The data appear to be underpowered for use of an 8-group ANOVA given the small number of data points per group, and potentially significant non-linearities in the number of transients over time are therefore ignored. Similarly, there appears to be a peak in the number of cells imaged at P9, but no details of which statistical testing was used are provided and again, more rigorous statistical approaches are required to understand whether significant parameter changes are occurring over time.Author Response – See response 3b above.Reviewer Response – See reviewer response to 3b above.

See Re-response to point #8.

11. Initial Review – In Figure 4, GABAergic axon activity was determined at P9-10 only, but this was interpreted as an increase. Without comparison to other ages, this conclusion would need to be tempered.Author Response – As explained above, there are few, if any, GABAergic axons in the CA1 pyramidal layer before P9 (Figure 4 —figure supplement 1 specifically, supplementary video 7). There is no signal, thus comparison is pointless. We have now clarified this point (lines 242-244), thanks.Reviewer Response – Thanks for clarifying this issue. To ensure that no pointless comparisons are inferred by the reader, it may be of benefit to change the wording of lines 255-256 from "increase in functional perisomatic GABAergic activity" to "emergence of functional perisomatic GABAergic activity."

We changed the manuscript accordingly see lines 259-260.

12. Initial Review – Statistical evaluation of data in Figure 4B also does not directly support the authors' conclusions. They explain in the results text a sharp increase in labeling at P9, but their statistical testing does not show a significant difference between P7 and P9, which is the supposed inflection point. This point further illustrates the need for more rigorous and sensitive statistical testing for age-dependent changes.Author Response – We had observed a significant difference between the P7 and P11 age groups (2 age groups exactly centered around P9), which demonstrates an increase in PV coverage around P9. In fact, P9 being the inflection point it is not surprising that it is neither different from P7 nor from P11 since it is precisely when axonal arborization is changing the most.Reviewer Response – The statistical interpretation is still problematic here. If the statistical comparison between the P7 and the P9 groups is not significant, it is not accurate to report in the results that "However, around P9, a sudden increase in the density of positive labeling was observed" (lines 240-241). The reasoning about P9 being an inflection point may be the case, but the currently employed statistics do not show it. If the authors wanted to show there is an inflection point, rather than just stating what their statistics show (a significant difference between P7 and P11), a modeling approach that demonstrates a change in the curvature of the modeled fit around P9 would be appropriate.

We changed the manuscript accordingly. See lines 245-247.

Reviewer #2 (Recommendations for the authors):Overall the authors have done a good job of responding to many of the points raised. However, I feel they have not provided a complete answer to three key points, raised by both myself and reviewer 1.1) Statistics for PMTHs. Thanks to the authors for clarifying that there was already shuffled significance criteria in place for the PMTHs and the addition of an asterisk to the plots helps to clarify this. However, I still do not feel that the explanation as it stands is fully satisfactory.a) It is necessary to provide more details of the shuffling procedure. The methods state simply that '500 surrogate raster plots per 770 imaging session were computed', which is insufficient detail. Unless the authors can provide a good reason otherwise, I think that the surrogates should be 20sec sections of data, drawn at random from the entire dataset, from both movement and non-movement epochs. If surrogates are not defined as above, the authors should justify why. In any case, more detail should be provided in the methods.

We would like to thank reviewer 2 for pointing out the lack of details present in the method section. To obtain a surrogate raster plot, the activity of each cell was translated by a randomly selected integer (between 1 and the total number of frames). This procedure has two consequences: it conserves the inter-transient time interval distribution y for each cell while disorganizing population activity.

Chance level PMTHs were built from these surrogate raster plots using the timestamps of movement onsets observed in the data. This allowed us to test the hypothesis that movement triggers a population response by synchronizing CA1 hippocampal cells. The reviewer proposes to use 20sec sections of data, drawn at random from the entire dataset. To our understanding this would be equivalent to selecting random movement onsets and conserving population activity. This would thus test a different hypothesis, namely that population synchrony could happen independently from movement and that, by chance, movement would precede SCEs. This was already tested at single-cell level in Figure 2 —figure supplement 1E where we found that 60% of the cells recorded at P5 are more active than expected by chance during movement epochs. We have now clarified the method section accordingly (see lines 769-771)

b) Are the 95th and 5th percentiles calculated with respect to each bin, or the entire 20s PMTH? If the former, then the authors need to account for the problem of multiple comparisons across the multiple time bins of the PMTH.

We would like to thank reviewer 2 for pointing out the lack of details present in the method section. To obtain a surrogate raster plot, the activity of each cell was translated by a randomly selected integer (between 1 and the total number of frames). This procedure has two consequences: it conserves the inter-transient time interval distribution y for each cell while disorganizing population activity.

Chance level PMTHs were built from these surrogate raster plots using the timestamps of movement onsets observed in the data. This allowed us to test the hypothesis that movement triggers a population response by synchronizing CA1 hippocampal cells. The reviewer proposes to use 20sec sections of data, drawn at random from the entire dataset. To our understanding this would be equivalent to selecting random movement onsets and conserving population activity. This would thus test a different hypothesis, namely that population synchrony could happen independently from movement and that, by chance, movement would precede SCEs. This was already tested at single-cell level in Figure 2 —figure supplement 1E where we found that 60% of the cells recorded at P5 are more active than expected by chance during movement epochs. We have now clarified the method section accordingly (see lines 769-771).

2) Statistics in addition to the PMTHs. Both Reviewer 1 and myself requested additional statistical analyses of the %cell activation data, in particular comparisons of how activation peaks and inhibition troughs evolve over development. The current argument of the authors is that it is sufficient to rely on whether PMTH activity crosses a significance threshold, or not, as a form of cross-age comparison. I don't agree with this – to take an example, the movement-activated activity peaks at P8 and P10 look visually very similar (Figure 2A). The proper statistical approach to test whether they are different is to compare them directly. Using the author's current approach, two very similar samples, not significantly different in themselves, could be judged to be from different populations based on one crossing an arbitrary shuffling-based threshold, and the other not.

We gently disagree with reviewer 2. As quantified in Figure 2B, the post movement activity in P8 mouse pups is significantly higher than the one in P10 mice. This is further confirmed by Figure 2C showing that most movements at P10 are followed by a reduction of activity whereas most movements at P8 are followed by an increase of activity. This allows us to directly compare the PMTHs presented in Figure 2A. Overall, even if PMTHs at P8 and P10 seem visually very similar to the reviewer, these two quantifications (Figure 2 B,C) show that they are not. In fact P10 PMTH is visually more similar to the one observed at P11 than to the P8, which is confirmed by Figures 2 B,C. Of note, reviewer #1 is now convinced by all of the changes made (see point 8 above).

The authors already have a model for better analysis in the manuscript – the analysis of activity troughs in Figure 2B. I think that a proper analysis of the data requires this method to be applied to movement-related peaks and post-movement throughs, for all data in figures 2, 3, and 4.

This method is already applied to all PMTHs included in Figure 2A and cannot be directly applied to Figure 4D that uses DF / F signal and not a percentage of active cells. Figure 3 shows the PMTHs for interneurons and pyramidal cells before P9 (P5-8) and after (P1012). The quantification presented in Figure 2B based on Figure 2A data was useful to quantify subtle differences in peaks and troughs between PMTHs (such as the P8 vs P10 example pointed out by the reviewer). In Figure 3, among the four PMTHs, three (interneurons P5-8 and P10-12 and pyramidal cells P5-8) show a significant increase and 1 (pyramidal cells P10-12) a significant decrease. Our only claim is that the activity of the two neuronal populations evolves differentially with respect to movement. Because these two very different PMTHs types cross the statistical threshold in opposite directions we do not think the data requires the Figure 2B method to be applied to movement-related peaks and post-movement throughs in this case (in agreement with reviewer #1 comment, see point #8). We agree with reviewer # 2 that quantification of peaks with this method could be useful to directly compare interneurons response to movement in P5-8 with P10-12 but this claim is outside the scope of the present study.

3) Thanks to the clarification from the authors, I now understand and accept that there are no axon terminals to image before P9. However, some more temporal precision regarding the emergence of perisomatic axon terminal activity would be helpful. The key transition dates for pyramidal cell activity run from P8 (still immature), P9 (which is transitional – no activity peak but also no trough) to P10 (mature activity). How does the emergence of axonal activity relate to this timeline? Is there a difference between P9 and P10? Is the response already mature at its first emergence (at P9)? Or does it continue to gradually increase between P9 and P12? This information would help make a more specific link between increases in perisomatic inhibition and PC activity.Unless the authors can show these data, then phrases such as 'functional surge' should be avoided ('surge' implies a rapid maturation, which cannot the demonstrated using one time point), the authors should restrict their conclusions to stating that 'functional perisomatic activity from inhibitory interneurons can be observed at P9-10 (or similar).

We have changed the manuscript accordingly see lines 259-260.